# Inhibiting IRE1α-endonuclease activity decreases tumor burden in a mouse model for hepatocellular carcinoma

**Nataša Pavlović[1], Carlemi Calitz[1], Kess Thanapirom[2], Guiseppe Mazza[2], Krista Rombouts[2], Pär Gerwins[1,3], Femke Heindryckx[1]***

[1]Department of Medical Cell Biology, Uppsala University, Uppsala, Sweden; [2]Regenerative Medicine & Fibrosis Group, Institute for Liver and Digestive Health, University College London, London, United Kingdom; [3]Department of Radiology, Uppsala University Hospital, Uppsala, Sweden

**Abstract** Hepatocellular carcinoma (HCC) is a liver tumor that usually arises in patients with cirrhosis. Hepatic stellate cells are key players in the progression of HCC, as they create a fibrotic micro-environment and produce growth factors and cytokines that enhance tumor cell proliferation and migration. We assessed the role of endoplasmic reticulum (ER) stress in the cross-talk between stellate cells and HCC cells. Mice with a fibrotic HCC were treated with the IRE1α-inhibitor 4μ8C, which reduced tumor burden and collagen deposition. By co-culturing HCC-cells with stellate cells, we found that HCC-cells activate IREα in stellate cells, thereby contributing to their activation. Inhibiting IRE1α blocked stellate cell activation, which then decreased proliferation and migration of tumor cells in different in vitro 2D and 3D co-cultures. In addition, we also observed cell-line-specific direct effects of inhibiting IRE1α in tumor cells.

## Introduction

***For correspondence:**
femke.heindryckx@mcb.uu.se

**Competing interests:** The authors declare that no competing interests exist.

Hepatocellular carcinoma (HCC) is a primary liver tumor that typically arises in a background of chronic liver disease and cirrhosis (*Calderaro et al., 2019*). One of the key players in the progression of cirrhosis to HCC is the hepatic stellate cell, which is activated during liver damage and differentiates towards a contractile myofibroblast-like cell that deposits extracellular matrix proteins (ECM), such as collagen (*Coulouarn and Clément, 2014*). Activated stellate cells can induce phenotypic changes in cancer cells through the production of growth factors and cytokines that stimulate tumor cell proliferation and induce a pro-metastatic phenotype (*Yu et al., 2013*). Malignant hepatocytes secrete high levels of transforming growth factor beta (TGFβ), which can contribute to the activation of stellate cells in the nearby stroma (*Giannelli et al., 2014*; *Nitta et al., 2008*; *Dooley et al., 2009*). These activated stellate cells are then responsible for the deposition of ECM. Several of the ECM-components such as proteoglycans, collagens, laminin, and fibronectin interact with tumor cells and cells in the stroma, which can directly promote cellular transformation and metastasis (*Lin et al., 2014*; *Song et al., 2016*). The ECM can also act as a reservoir for growth factors and cytokines, which can be rapidly released to support the tumor's needs. In addition, activated stellate cells contribute to a highly vascularized tumor micro-environment, by secreting pro-angiogenic molecules and by recruiting pro-angiogenic (and pro-tumoral) myeloid and lymphoid derived cell types (*Zhang et al., 2017*). By constricting the hepatic microvasculature, they also cause hypoxia, which contributes to the angiogenic switch and can induce a more aggressive tumor phenotype (*Taura et al., 2008*). It is therefore not surprising that tumor cells actively secrete growth factors (such as TGFβ) to induce activation and migration of stellate cells, which creates a fibrotic environment that further supports and enhances tumor progression (*Coulouarn and Clément, 2014*;

*Caja et al., 2018*; *Lu et al., 2015*). Since activated stellate cells play an essential role in the onset and progression of HCC, blocking their activation has been proposed as a potential therapy for patients with HCC (*Carloni et al., 2014*). One strategy to prevent stellate cell activation, is by blocking the IRE1α-pathway of the unfolded protein response (UPR) (*Heindryckx et al., 2016*; *Liu et al., 2019*).

The UPR serves to cope with the accumulation of misfolded or unfolded proteins in the endoplasmic reticulum (ER) in an attempt to restore protein folding, increase ER-biosynthetic machinery and maintain cellular homeostasis (*Schröder and Kaufman, 2005*). It can exert a cytoprotective effect by re-establishing cellular homeostasis, while apoptotic signaling pathways will be activated in case of severe and/or prolonged ER-stress (*Lam et al., 2020*). The presence of misfolded proteins is sensed via three transmembrane proteins in the ER: inositol requiring enzyme 1α (IRE1α), protein kinase RNA-like ER-kinase (PERK) and activating transcription factor 6α (ATF6α) (*Acosta-Alvear et al., 2018*). The development of solid tumors is characterized by uncontrolled growth and proliferation of malignant cells, resulting in a compact mass of cells and a hypoxic tumor micro-environment, two conditions that are well-characterized ER-stress inducers. Therefore, it is not surprising that activation of the UPR represents a major hallmark of several solid tumors, such as breast cancer (*Liang et al., 2018*), colon cancer (*Li et al., 2017b*), and HCC (*Vandewynckel et al., 2013*). The induction of the UPR in cancer cells may serve as a double-edged sword, which can aid tumor progression as well as prevent tumor growth in a context-dependent manner. Persistent ER-stress can activate pathways that induce cell death, effectively eliminating cells with a potential to become malignant. On the other hand, tumor cells may hijack the ER-stress pathways to provide survival signals required for uncontrolled growth and eventually avoid apoptosis (*Kim et al., 2015*). Activation of the UPR has also been shown to affect different fibrotic diseases (*Heindryckx and Li, 2018*), including non-alcoholic fatty liver disease (*Bandla et al., 2018*; *Kwanten et al., 2016*; *Dasgupta et al., 2020*), hepatitis-B-induced carcinogenesis (*Li et al., 2017a*), and biliary cirrhosis (*Sasaki et al., 2015*). We have previously shown that inhibiting the IRE1α-branch of the UPR-pathway using 4μ8C, blocks TGFβ-induced activation of fibroblasts and stellate cells in vitro and reduces liver fibrosis in vivo (*Heindryckx et al., 2016*). In the current study, our aim was to define the role of IRE1α in the cross-talk between hepatic stellate cells and tumor cells in liver cancer. We show that pharmacologic inhibition of the IRE1α-signaling pathway decreases tumor burden in a chemically induced mouse model for HCC. Using several in vitro co-culturing methods, we identified that blocking IRE1α in hepatic stellate cells prevents their activation. This then decreases proliferation and migration of tumor cells in co-cultures, in addition to the direct effect of inhibiting IRE1α in tumor cells. Our results also indicate that there are cell-line-specific differences in how cells respond to IRE1α-inhibition, including differences in the IRE1α-dependent generation of reactive oxygen species.

## Results

### Pharmacological inhibition of IRE1α reduces tumor burden in a chemically induced mouse model for HCC

Hepatocellular carcinoma was induced in mice by weekly injections with N-nitrosodiethylamine (DEN) for 25 weeks (*Heindryckx et al., 2010*). From week 10, IRE1α-endonuclease activity was pharmacologically inhibited with 4μ8C. Histological analysis of liver tissue confirmed the presence of liver tumors in a fibrotic background at 25 weeks (*Figure 1A*). Treatment with 4μ8C significantly reduced tumor burden (*Figure 1B*), as measured on H and E-stained liver sections (*Figure 1A*). Stellate cell activation and liver fibrosis was quantified by Sirius Red staining (*Figure 1A and C*) and immunohistochemical staining with αSMA-antibodies (*Figure 1A and D*) on liver sections. Mice with HCC had a significant increase in the percentage of collagen (*Figure 1C*) and αSMA-staining (*Figure 1D*), compared to healthy mice. Treatment with 4μ8C restored collagen (*Figure 1C*) and αSMA-levels (*Figure 1D* and *Figure 1E*) to healthy baseline levels. mRNA-expression levels of *Pcna* were determined on tumor nodules and surrounding non-tumor stromal tissue (*Figure 1E*). As expected, proliferation of cells was increased within the tumor itself, compared to the levels in healthy liver tissue and stromal tissue. Treatment with 4μ8C significantly decreased the levels of *Pcna*-mRNA expression within the tumor, suggesting a decrease in tumor cell proliferation. A proteomics array using the

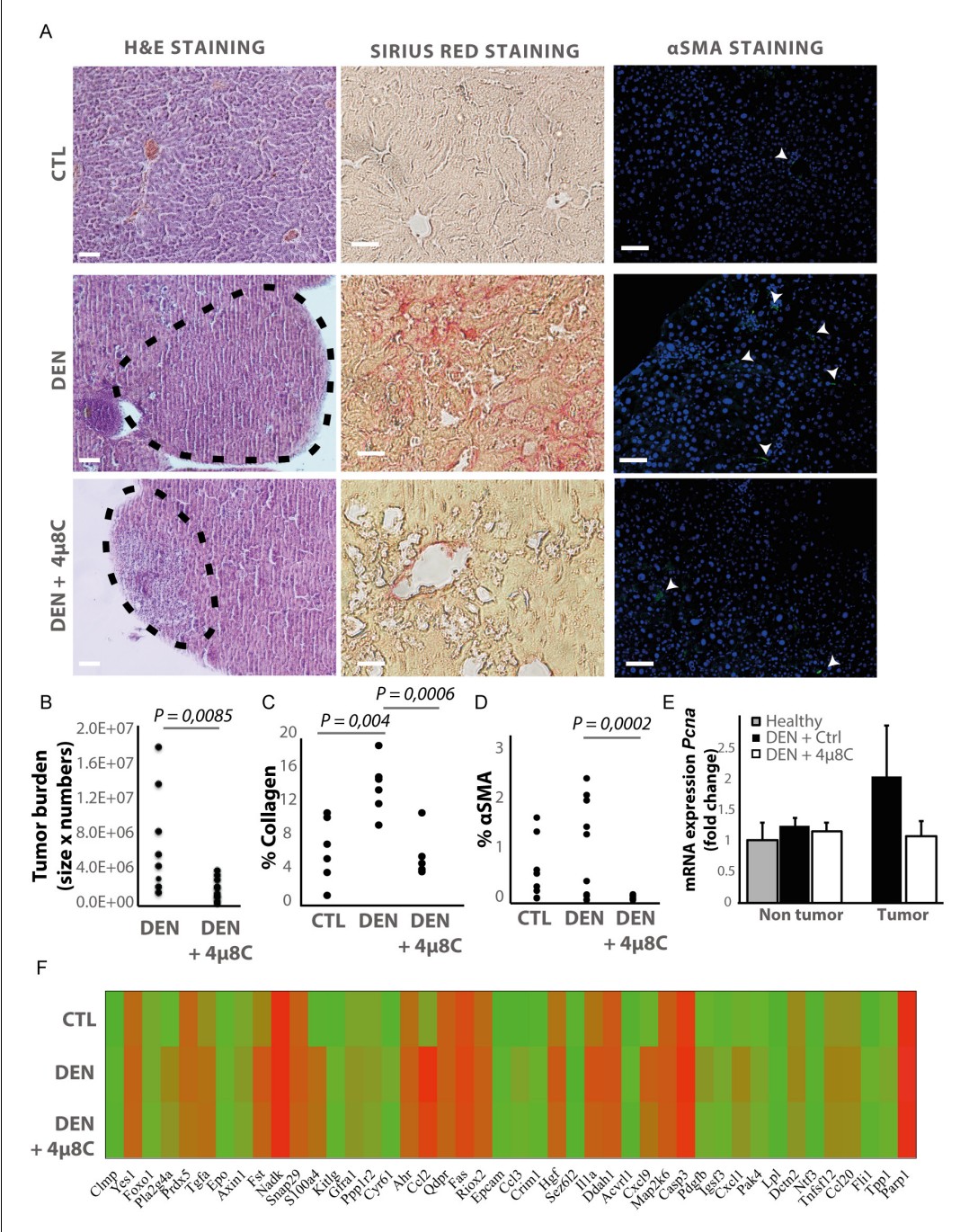

**Figure 1.** Inhibiting IRE1α reduces tumor burden in vivo. (A) Representative images of liver slides stained with hematoxylin and eosin (H and E), Sirius red and αSMA-antibodies. (B) tumor burden of mice with DEN-induced HCC treated with 4μ8C or vehicle-treated controls. (C) Quantification of percentage of collagen and (D) αSMA on liver slides. (E) mRNA expression of *Pcna* in liver tissue from mice with HCC treated with 4μ8C (F). Heatmap showing protein expression levels in healthy liver, DEN-induced HCC and DEN-induced HCC treated with 4μ8C from three biological replicates per group. p-Values were calculated via the Student's T-test, scale bars = 120 μm.

Olink Mouse Exploratory assay revealed that DEN-induced murine tumors had a significantly increased protein expression of 20 oncogenic proteins compared to healthy controls (*Figure 1F* and *Table 1*). In the 4μ8C-treated group, only 11 oncogenic proteins were increased compared to healthy controls (*Figure 1F* and *Table 1*). Treatment with 4μ8C also significantly reduced protein expression of two HCC promotors, PRDX5, and DDAH1 (*Figure 1F* and *Table 1*).

**Table 1.** A proteomics array using the Olink Mouse Exploratory assay – source data *Figure 1F*.

| Protein name | Biological process | CTL Mean | St. Dev | Den Average | St. Dev | DEN+4 u8c Average | St. Dev | DEN vs Ctrl | DEN vs 4 u8C | Ctrl vs 4 u8C |
|---|---|---|---|---|---|---|---|---|---|---|
| Clmp | Not prognostic in HCC | 1.68 | 0.14 | 2.97 | 1.00 | 2.48 | 0.64 | * | | |
| Yes1 | HCC promotor | 7.11 | 0.29 | 7.51 | 0.20 | 7.44 | 0.19 | * | | |
| Foxo1 | Tumor suppressor | 4.15 | 0.06 | 4.12 | 0.73 | 3.87 | 0.49 | | | |
| Pla2g4a | HCC promotor | 3.42 | 0.38 | 5.70 | 1.36 | 5.04 | 0.80 | * | | * |
| Prdx5 | HCC promotor | 7.37 | 0.49 | 7.23 | 0.26 | 6.67 | 0.34 | | * | |
| Tgfa | Tumor growth factor | 5.36 | 0.52 | 6.81 | 0.64 | 6.93 | 0.88 | * | | * |
| Epo | Unfavorable prognotic marker | 3.20 | 0.34 | 3.71 | 0.35 | 3.37 | 0.33 | | | |
| Axin1 | HCC promotor | 4.24 | 0.38 | 4.80 | 0.37 | 4.39 | 0.35 | | | |
| Fst | HCC promotor | 5.87 | 0.31 | 8.04 | 0.73 | 7.50 | 0.71 | * | | * |
| Nadk | Not prognostic in HCC | 10.10 | 0.13 | 10.14 | 0.18 | 10.30 | 0.27 | | | |
| Snap29 | Not prognostic in HCC | 7.70 | 0.32 | 7.87 | 0.32 | 7.62 | 0.30 | | | |
| S100a4 | HCC promotor | 2.73 | 0.74 | 7.01 | 0.62 | 6.85 | 0.97 | * | | * |
| Kitlg | Metastasis | 2.48 | 0.42 | 3.74 | 0.62 | 3.31 | 0.98 | * | | |
| Gfra1 | HCC promotor | 4.40 | 0.35 | 5.07 | 0.40 | 4.92 | 0.39 | * | | |
| Ppp1r2 | Not prognostic in HCC | 4.37 | 0.16 | 4.86 | 0.46 | 4.47 | 0.43 | | | |
| Cyr61 | HCC promotor | 2.40 | 0.53 | 4.14 | 1.64 | 3.13 | 1.22 | * | | |
| Ahr | Not prognostic in HCC | 6.95 | 0.46 | 7.68 | 0.74 | 7.38 | 0.64 | | | |
| Ccl2 | HCC promotor | 4.59 | 0.58 | 9.69 | 2.04 | 8.93 | . | * | | * |
| Qdpr | Not prognostic in HCC | 7.71 | 0.11 | 7.72 | 0.14 | 7.54 | 0.15 | | | |
| Fas | HCC promotor | 8.66 | 0.18 | 8.83 | 0.18 | 8.70 | 0.18 | | | |
| Riox2 | HCC promotor | 7.10 | 0.15 | 7.71 | 0.38 | 7.59 | 0.14 | * | | * |
| Epcam | HCC promotor | 1.56 | 0.33 | 3.16 | 1.14 | 3.27 | 0.89 | * | | |
| Ccl3 | Prognostic marker | 1.49 | 0.39 | 4.42 | 1.86 | 3.73 | 1.07 | * | | * |
| Crim1 | HCC promotor | 2.46 | 0.28 | 3.71 | 1.09 | 3.21 | 0.56 | * | | * |
| Hgf | Tumor growth factor | 6.69 | 0.35 | 7.94 | 1.01 | 7.41 | 0.71 | * | | |
| Sez6l2 | HCC promotor | −0.29 | 0.15 | 0.61 | 0.53 | 0.19 | 0.29 | * | | |
| Il1a | Inflammation and fibrosis | 6.65 | 0.51 | 8.35 | 0.65 | 7.62 | 0.54 | * | | * |
| Ddah1 | HCC promotor | 8.04 | 0.22 | 8.18 | 0.05 | 7.84 | 0.18 | | * | |
| Acvrl1 | Not prognostic in HCC | 2.09 | 0.18 | 3.44 | 1.31 | 2.81 | 0.47 | | | |
| Cxcl9 | Inflammation and fibrosis | 3.68 | 0.86 | 7.71 | 1.68 | 6.65 | 1.58 | * | | * |
| Map2k6 | Not prognostic in HCC | 7.75 | 0.15 | 7.98 | 0.41 | 7.88 | 0.28 | | | |
| Casp3 | Tumor surrpressor | 9.22 | 0.19 | 9.74 | 0.35 | 9.43 | 0.26 | | | |
| Pdgfb | Tumor growth factor | 3.52 | 0.31 | 4.96 | 1.27 | 3.97 | 0.40 | * | | |
| Igsf3 | Unfavorable prognotic marker | 3.12 | 0.28 | 4.19 | 0.82 | 3.64 | 0.72 | | | |
| Cxcl1 | HCC promotor | 3.77 | 0.40 | 5.74 | 0.78 | 5.06 | 0.51 | * | | * |
| Pak4 | HCC promotor | 3.47 | 0.42 | 4.39 | 0.68 | 3.93 | 0.54 | | | |
| Lpl | Not prognostic in HCC | 1.66 | 0.40 | 2.44 | 0.45 | 2.02 | 0.60 | | | |
| Dctn2 | Unfavorable prognotic marker | 5.48 | 1.31 | 5.67 | 0.70 | 4.98 | 0.55 | | | |
| Ntf3 | Not prognostic in HCC | 2.16 | 0.27 | 2.80 | 0.71 | 2.27 | 0.40 | | | |
| Tnfsf12 | HCC promotor | 5.28 | 0.35 | 6.00 | 0.76 | 5.59 | 0.62 | | | |
| Ccl20 | Unfavorable prognotic marker | 5.20 | 0.34 | 5.92 | 0.81 | 5.53 | 0.66 | | | |
| Fli1 | HCC promotor | 1.91 | 0.22 | 3.73 | 1.38 | 2.98 | 0.83 | | | |
| Tpp1 | Unfavorable prognotic marker | 3.67 | 0.38 | 4.24 | 0.64 | 3.73 | 0.50 | | | |
| Parp1 | Unfavorable prognotic marker | 10.30 | 0.72 | 10.93 | 0.49 | 10.51 | 0.62 | | | |

## Markers of the unfolded protein response are upregulated in HCC and mainly located in the tumor stroma

mRNA-levels of different ER-stress-genes were measured in tumor and surrounding non-tumor tissue of mice with DEN-induced HCC (*Figure 2A*). *Hspa5*-mRNA-expression was increased in the surrounding non-tumor tissue of DEN-induced mice with HCC, while there was no difference within the tumor, compared to healthy controls (*Figure 2A and B*). Western blot confirmed the increase of BIP-protein expression in DEN-induced livers, which was reduced after treatment with 4μ8C (*Figure 2C*). The ratio of spliced to unspliced *Xbp1*-mRNA was significantly increased in the surrounding non-tumor tissue of DEN-induced mice (*Figure 2D*). Treatment with 4μ8C significantly reduced the ratio of spliced to unspliced *Xbp1*-mRNA in surrounding non-tumorous stromal tissue (*Figure 2D*). Western blot on whole tissue samples – containing both tumor and non-tumoral tissue – also confirmed a significant decrease of XBP1-splicing after treatment with 4μ8C (*Figure 2E,F and G*). Immunohistochemical straining with XBP1-antibodies against the spliced variant further demonstrate that the expression of spliced XBP1 is mainly located in the peritumoral area (*Figure 2H*). Spliced XBP1 was significantly increased in the DEN-induced liver tissue and treatment with 4μ8C restored these levels to a similar level as seen in healthy controls (*Figure 2I*). Co-staining of liver tissue with antibodies against αSMA and antibodies against spliced XBP1 (*Figure 2—figure supplements 1A* and *2A*), total XBP1 (*Figure 2—figure supplements 1B* and *2B*), IRE1α (*Figure 2—figure supplements 1C* and *2C*), phospho-IRE1α (*Figure 2—figure supplements 1D* and *2D*), and BIP (*Figure 2—figure supplements 1E* and *2E*), revealed that expression of markers from the IRE1α-pathway were mainly localized within activated stellate cells in the liver, although other hepatic cell populations also expressed some of these markers. At a higher magnification (*Figure 2—figure supplement 1F*), it also becomes clear that the expression of spliced XBP1 is not only cytoplasmic but some staining appears peri-nuclear and nuclear.

A gene-set enrichment assay on microarray data from HCC-patients with fibrotic septae and without fibrotic septae showed an increase of genes involved in the UPR in the fibrotic HCC samples compared to non-fibrous HCC (*Figure 3A*). Several actors of the IRE1α-branch of the UPR are amongst the genes that contribute to the core-enrichment of this analysis (*Table 2*). Immunohistochemical staining of liver biopsies from HCC-patients further confirmed presence of IRE1α-mediated ER-stress markers *BIP*, *PPP2R5B*, *SHC1*, and WIPI1 localized in the fibrotic scar tissue and near hepatic blood vessels (*Figure 3B*). In addition, increased expression of these markers was significantly correlated with poor survival in patients with liver cancer (*Figure 3C*).

## Tumor cells secrete factors that induce ER-stress in hepatic stellate cells

Hepatic stellate cell-lines (LX2) and HCC-cell lines (HepG2 and Huh7) were grown in different compartments using a transwell-assay. This confirmed that tumor cells secrete factors that induce mRNA-expression of *EIF2AK3*, *DDIT3*, *HSPA5* (*Figure 4A*), spliced *XBP1* (*Figure 4B,C and D*), and *HSPA5* (*Figure 4C*), as well as protein expression of p-IRE1α (*Figure 4F*) in hepatic stellate cells co-cultured with tumor cells, indicating the presence of ER-stress. Co-culturing also led to their activation, as measured by mRNA-expression of *ACTA2* (*Figure 4F*) and collagen (*Figure 4G*) in LX2-cells grown with HepG2 or Huh7-cells in a transwell-assay. The mRNA-expression of *ACTA2* and collagen was restored to baseline levels when 4μ8C was added to the transwell co-cultures.

De-cellularized human liver 3D-scaffolds were engrafted with hepatic stellate cells (LX2) and tumor cells (HepG2). Sirius red staining and H and E staining confirmed that that LX2-cells and HepG2-cells successfully engrafted the collagen-rich matrix of the decellularized human liver scaffolds (*Figure 5A and B*). Engrafting both LX2-stellate cells and HepG2-cancer cells led to a significant increase of collagen staining (*Figure 5B*) and mRNA-expression of collagen, *HSPA5*, and spliced *XBP1* (*Figure 5C*) compared to scaffolds that were only engrafted with LX2-cells. Adding 4μ8C significantly decreased mRNA-expression of collagen and *HSPA5* in the LX2 and HepG2 co-cultured scaffolds (*Figure 5C*).

Tumor cells are important sources of TGFβ, which is a known activator of stellate cells. Surprisingly, measuring TGFβ in mono-cultures lead to undetectable levels of TGFβ in Huh7-cells and low-levels in HepG2-cells (*Figure 4—figure supplement 1A*). These levels increased when LX2-cells were added to the co-cultures (*Figure 4—figure supplement 1A*). Engrafting both LX2-stellate cells and HepG2-cancer cells in the human liver scaffolds, slightly increased TGFβ-levels in the medium

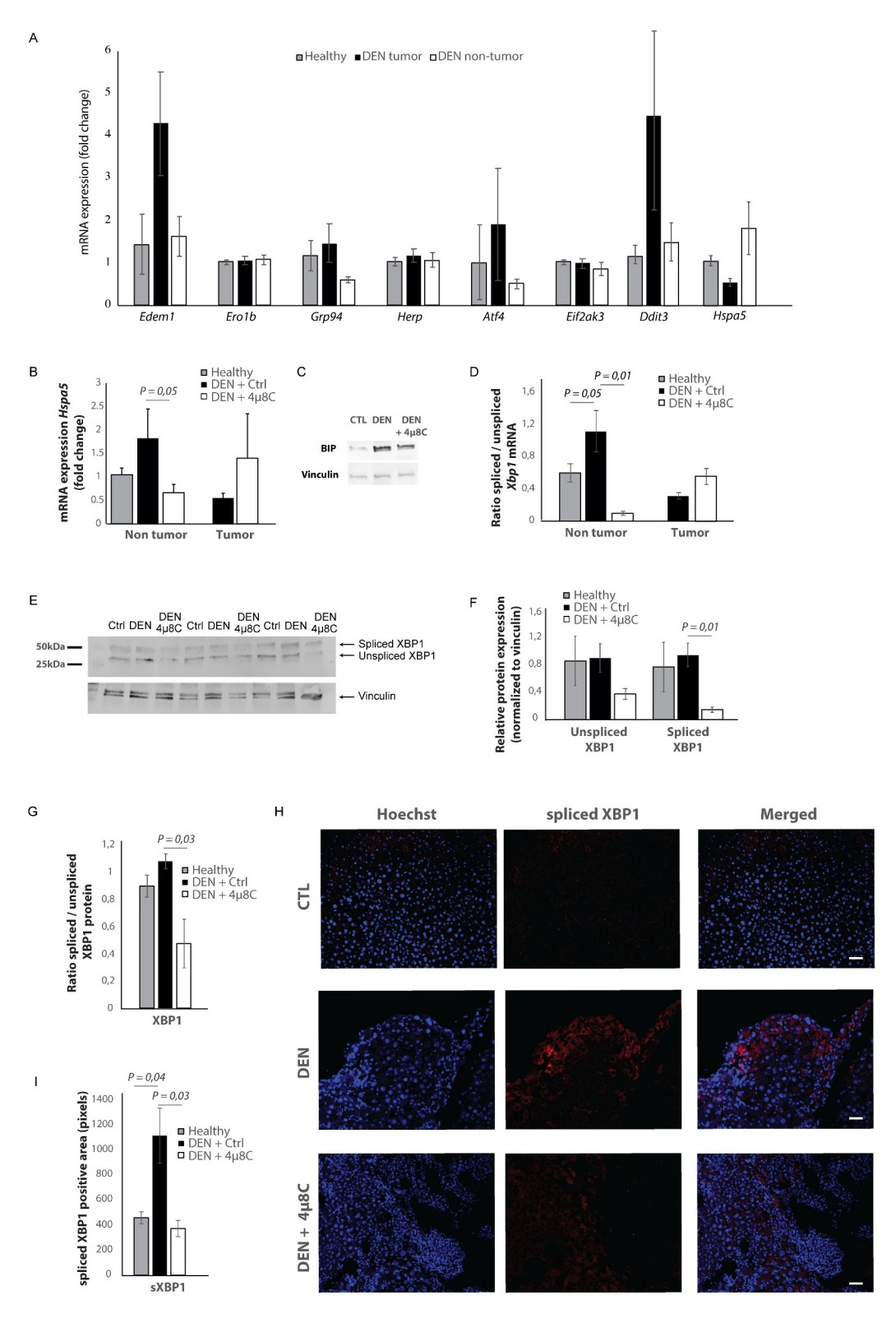

**Figure 2.** Increased expression of ER-stress markers in mice with HCC. (**A**) mRNA expression of ER-stress markers *Edem1*, *Ero1b*, *Grp94*, *Herp*, *Atf4*, *Eif2ak3*, *Ddit3*, and *Hspa5* in liver tissue from healthy mice; and tumor tissue and surrounding non-tumoral tissue from mice with DEN-induced HCC. (**B**) *Hspa5*-mRNA and (**C**) protein expression of BIP in murine liver tissue. (**D**) Ratio of spliced to unspliced *XBP1* in liver tissue from healthy mice; and tumor tissue and surrounding non-tumoral tissue from mice with DEN-induced HCC, treated with 4μ8C. (**E**) Representative western blot image of spliced and
*Figure 2 continued on next page*

*Figure 2 continued*

unspliced XBP1 protein and vinculin in healthy liver, DEN-induced HCC and DEN-induced HCC treated with 4µ8C. (**F**) quantification of spliced and unspliced XBP1, normalized to total vinculin levels. (**G**) Ratio of spliced to unspliced XBP1 protein levels. (**H**) Representative images and (**I**) quantification of liver tissue sections stained with antibodies against spliced XBP1. p-Values were calculated via the Student's T-test with five biological replicates per group. Scale bars = 120 µm.

The online version of this article includes the following figure supplement(s) for figure 2:

**Figure supplement 1.** Activation of the unfolded protein response is mainly located in the stroma of mice with HCC.

**Figure supplement 2.** Expression of ER-stress markers is localized in close vicinity to αSMA.

compared to scaffolds engrafted by only one cell type, but overall no significant differences were seen (*Figure 4—figure supplement 1B*). It is important to note that the baseline TGFβ-levels were markedly higher in the mono-cultured scaffolds, compared to the levels measured in cells grown in a standard 2D in vitro set-up (*Figure 4—figure supplement 1A*). Blocking TGFβ-receptor signaling with SB-431541 significantly reduced mRNA-expression of ER-stress markers *DDIT3* (*Figure 4—*

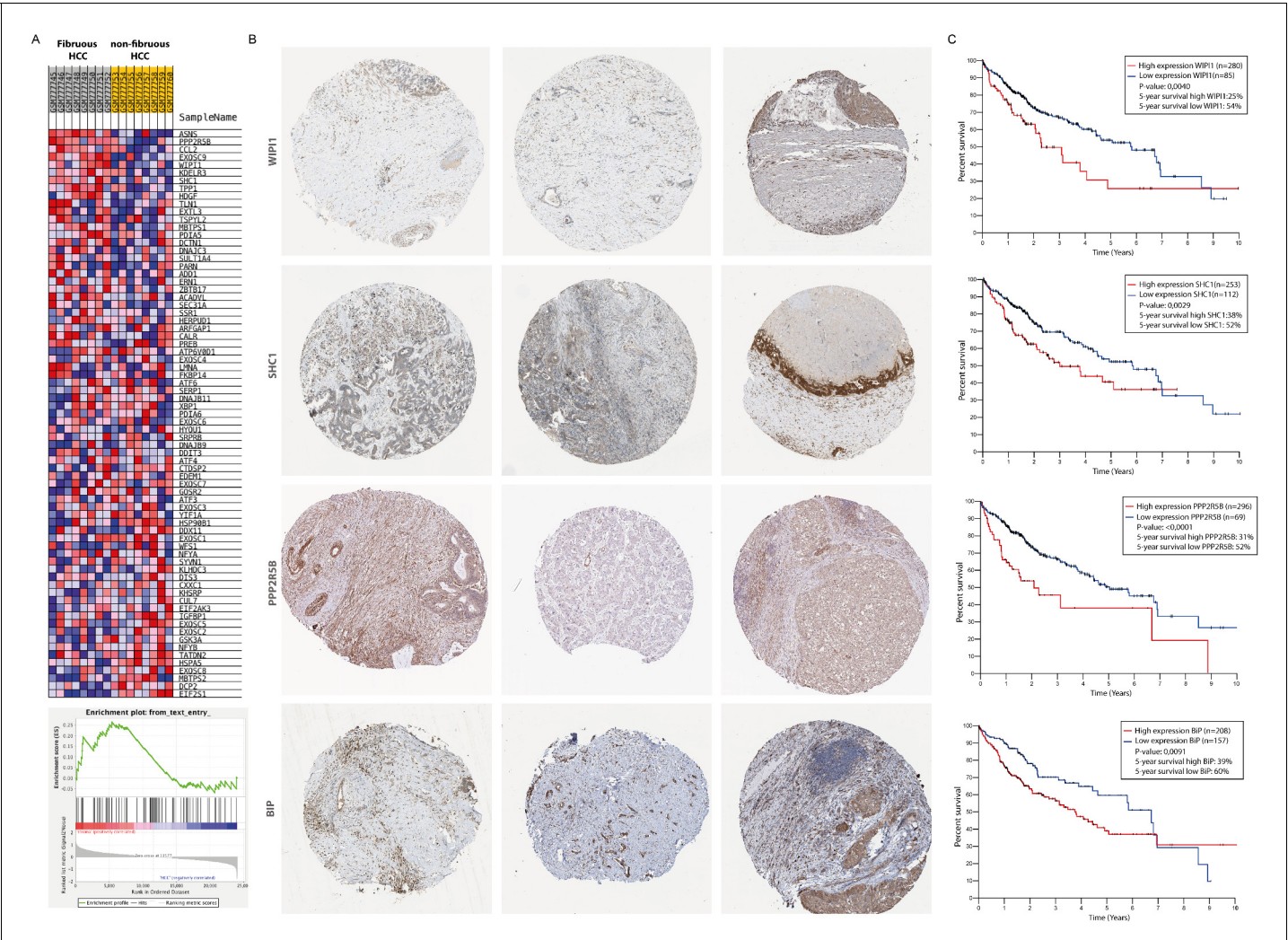

**Figure 3.** Activation of the unfolded protein response pathway is increased in patients with fibrotic HCC. (**A**) Heat map showing gene-set enrichment analysis results from samples from fibrous HCC versus non-fibrous HCC. (**C**) Immunohistochemically stained liver biopsies from HCC-patients obtained from the human protein atlas, using antibodies against IRE1α-mediated actors of the unfolded protein response: WIPI1, SHC1, PPP2R5B, and BIP. (**D**) Kaplan-Meier survival curves of HCC-patients with high or low expression of *WIPI1, SHC1, PPP2R5B,* and *BIP*. p-Values were calculated via a Log-Rank test.

**Table 2.** Genes the contributed to the core-enrichment of the GSEA.

| Probe | Description | Rank Gene list | Rank Metric score | Core enrichment | UPR branch |
|---|---|---|---|---|---|
| ASNS | Asparagine synthetase (glutamine-hydrolyzing) [Source:HGNC Symbol;Acc:HGNC:753] | 207 | 0.940 | Yes | Perk |
| PPP2R5B | Protein phosphatase two regulatory subunit B'beta [Source:HGNC Symbol;Acc:HGNC:9310] | 423 | 0.821 | Yes | Ire1a |
| CCL2 | C-C motif chemokine ligand 2 [Source:HGNC Symbol;Acc:HGNC:10618] | 847 | 0.689 | Yes | Ire1a and Perk |
| EXOSC9 | Exosome component 9 [Source:HGNC Symbol;Acc:HGNC:9137] | 1004 | 0.654 | Yes | Ire1a and Perk |
| WIPI1 | WD repeat domain, phosphoinositide interacting 1 [Source:HGNC Symbol;Acc:HGNC:25471] | 1022 | 0.649 | Yes | Ire1a |
| KDELR3 | KDEL endoplasmic reticulum protein retention receptor 3 [Source:HGNC Symbol;Acc:HGNC:6306] | 1106 | 0.635 | Yes | Ire1a |
| SHC1 | SHC adaptor protein 1 [Source:HGNC Symbol;Acc:HGNC:10840] | 2691 | 0.432 | Yes | Ire1a |
| TPP1 | Tripeptidyl peptidase 1 [Source:HGNC Symbol;Acc:HGNC:2073] | 2884 | 0.414 | Yes | Ire1a |
| HDGF | Heparin binding growth factor [Source:HGNC Symbol;Acc:HGNC:4856] | 3235 | 0.386 | Yes | Ire1a |
| TLN1 | Talin 1 [Source:HGNC Symbol;Acc:HGNC:11845] | 3264 | 0.384 | Yes | Ire1a |
| EXTL3 | Exostosin like glycosyltransferase 3 [Source:HGNC Symbol;Acc:HGNC:3518] | 3488 | 0.365 | Yes | Ire1a |
| TSPYL2 | TSPY like 2 [Source:HGNC Symbol;Acc:HGNC:24358] | 3680 | 0.350 | Yes | Ire1a |
| MBTPS1 | Membrane-bound transcription factor peptidase, site 1 [Source:HGNC Symbol;Acc:HGNC:15456] | 3996 | 0.327 | Yes | Atf6 |
| PDIA5 | Protein disulfide isomerase family A member 5 [Source:HGNC Symbol;Acc:HGNC:24811] | 4530 | 0.294 | Yes | Ire1a |
| DCTN1 | Dynactin subunit 1 [Source:HGNC Symbol;Acc:HGNC:2711] | 4638 | 0.287 | Yes | Ire1a |
| DNAJC3 | DnaJ heat-shock protein family (Hsp40) member C3 [Source:HGNC Symbol;Acc:HGNC:9439] | 4761 | 0.281 | Yes | Ire1a |
| SULT1A4 | Sulfotransferase family 1A member 4 [Source:HGNC Symbol;Acc:HGNC:30004] | 4938 | 0.272 | Yes | Ire1a |
| PARN | Poly(A)-specific ribonuclease [Source:HGNC Symbol;Acc:HGNC:8609] | 5037 | 0.266 | Yes | Perk |
| ADD1 | Adducin 1 [Source:HGNC Symbol;Acc:HGNC:243] | 5375 | 0.250 | Yes | Ire1a |
| ERN1 | Endoplasmic reticulum to nucleus signaling 1 [Source:HGNC Symbol;Acc:HGNC:3449] | 5411 | 0.248 | Yes | Ire1a |

figure supplement 1C), spliced *XBP1* (*Figure 4—figure supplement 1D*) and *HSPA5* (*Figure 4—figure supplement 1F*) in stellate cells co-cultured with tumor cells using transwells. Adding a TGFβ-receptor-inhibitor to stellate cell – tumor cell co-cultures also reduced stellate cell activation, as measured by mRNA-expression of *ACTA2* (*Figure 4—figure supplement 1G*) and collagen (*Figure 4—figure supplement 1H*). This indicates that TGFβ-secretion by tumor cells could be, at least in part, responsible for activating stellate cells and for inducing the IRE1α-branch of the UPR.

## Pharmacological inhibition of IRE1α decreases tumor cell proliferation in stellate cell – tumor cell co-cultures

In transwell co-culturing assays, we found that co-culturing HepG2 or Huh7-tumor cells with LX2-stellate cells significantly increased *PCNA*-mRNA-expression in HepG2 and Huh7-tumor cell lines (*Figure 6A*). Adding 4μ8C significantly decreased mRNA-expression of *PCNA* in Huh7-cells grown in a transwell co-culture with LX2-cells, while not affecting *PCNA*-expression in tumor cell monocultures (*Figure 6A*). *PCNA*-levels in HepG2-LX2 transwell co-cultures were slightly decreased, but this was not significant. Proliferation was measured 24 hr after exposure to 4μ8C in tumor cells (HepG2 and Huh7) grown as mono-cultures and in co-culture with LX2-stellate cells. While 4μ8C induced a significant increase in proliferation of HepG2-monocultures, no difference was seen in LX2-monocultures and a significant decrease was seen in the HepG2-LX2 co-cultures (*Figure 6B*). In the Huh7 tumor cell line, 4μ8C significantly decreased cell number compared to untreated controls and a similar

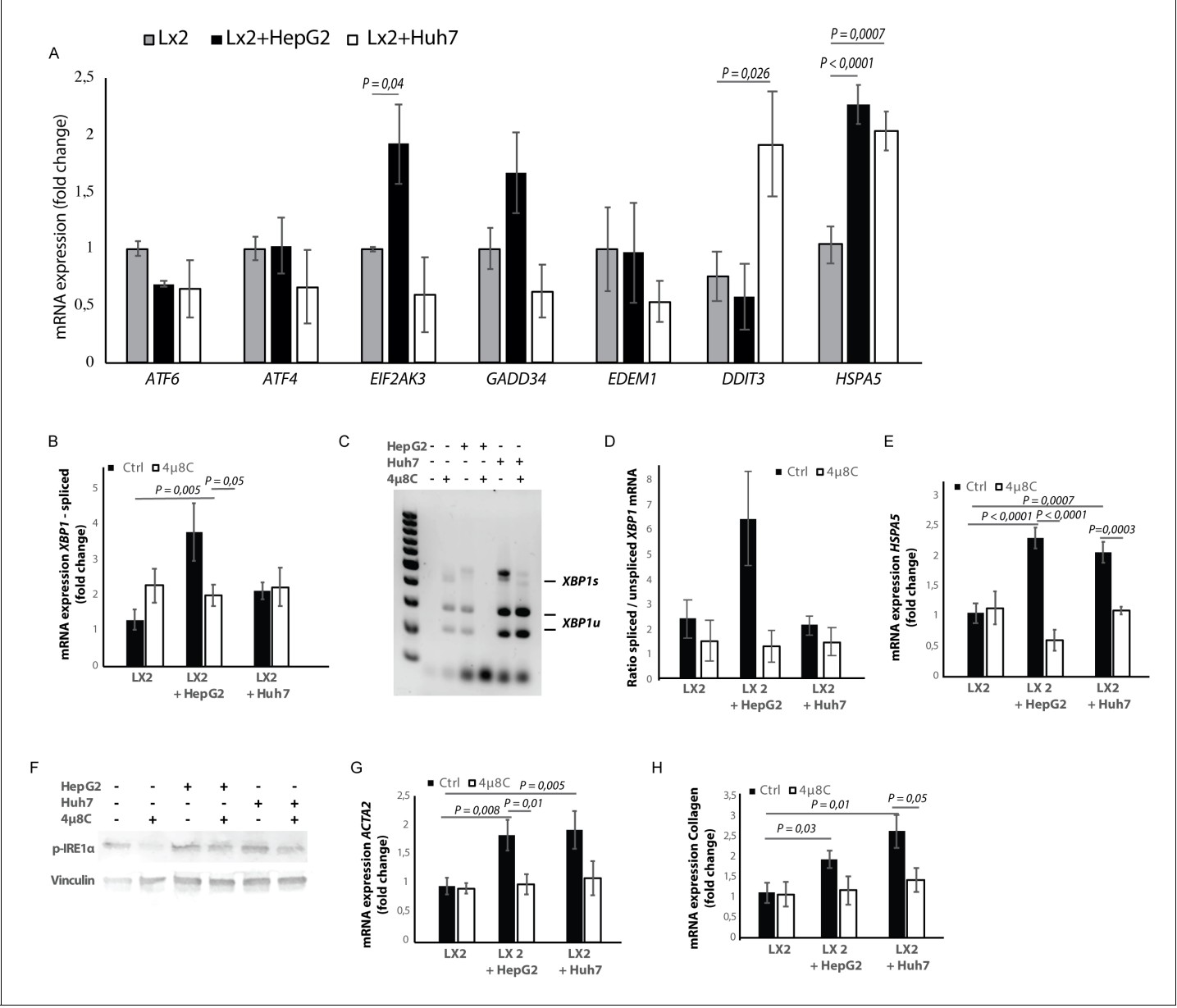

**Figure 4.** Tumor cells secrete factors that induce ER-stress in stellate cells, which contributes to their activation. (**A**) mRNA-expression of ER-stress markers *ATF6, ATF4, EIF2AK3, GADD34, EDEM1, DDIT3* and *HSPA5*, in stellate cells (LX2) co-cultured with cancer cells (HepG2 or Huh7) and treated with 4μ8C or control. (**B**) Detection of spliced (*XBP1s*) and unspliced *XBP1* (*XBP1u*) via qPCR and (**C**) via digestion of the XBP1u-RT-qPCR product by *Pst-I* and subsequent visualization by separation of on agarose gel. (**D**) Quantified ratio of spliced and unspliced measured on agarose gel after digestion by *Pst-I* (**E**) mRNA expression of HSPA5 in stellate cells (LX2) co-cultured with cancer cells (HepG2 or Huh7) and treated with 4μ8C or control. (**F**) protein expression of p-IRE1α and vinculin in stellate cells (LX2) co-cultured with cancer cells (HepG2 or Huh7) in transwell assays and treated with 4μ8C or control. (**G**) mRNA-expression of stellate cell activation markers *ACTA2* and (**H**) collagen in LX2-cells co-cultured with HepG2 or Huh7-cells and treated with or without 4μ8C. p-Values were calculated via ANOVA with 10 biological replicates per group.

The online version of this article includes the following figure supplement(s) for figure 4:

**Figure supplement 1.** Secretion of TGFβ by tumor cells activates stellate cells and induces ER-stress.

reduction was seen in the Huh7-LX2 co-cultures (**Figure 6C**). Immunohistochemical staining with anti-bodies against EPCAM and KI67 show that the effect on proliferation is mainly localized in the tumor cell population of these co-cultures (**Figure 6D**).

3D-spheroids were generated using tumor cells alone (HepG2 or Huh7) or in combination with LX2-cells. While the HepG2-spheroids experienced a lower proliferation rate when generated in

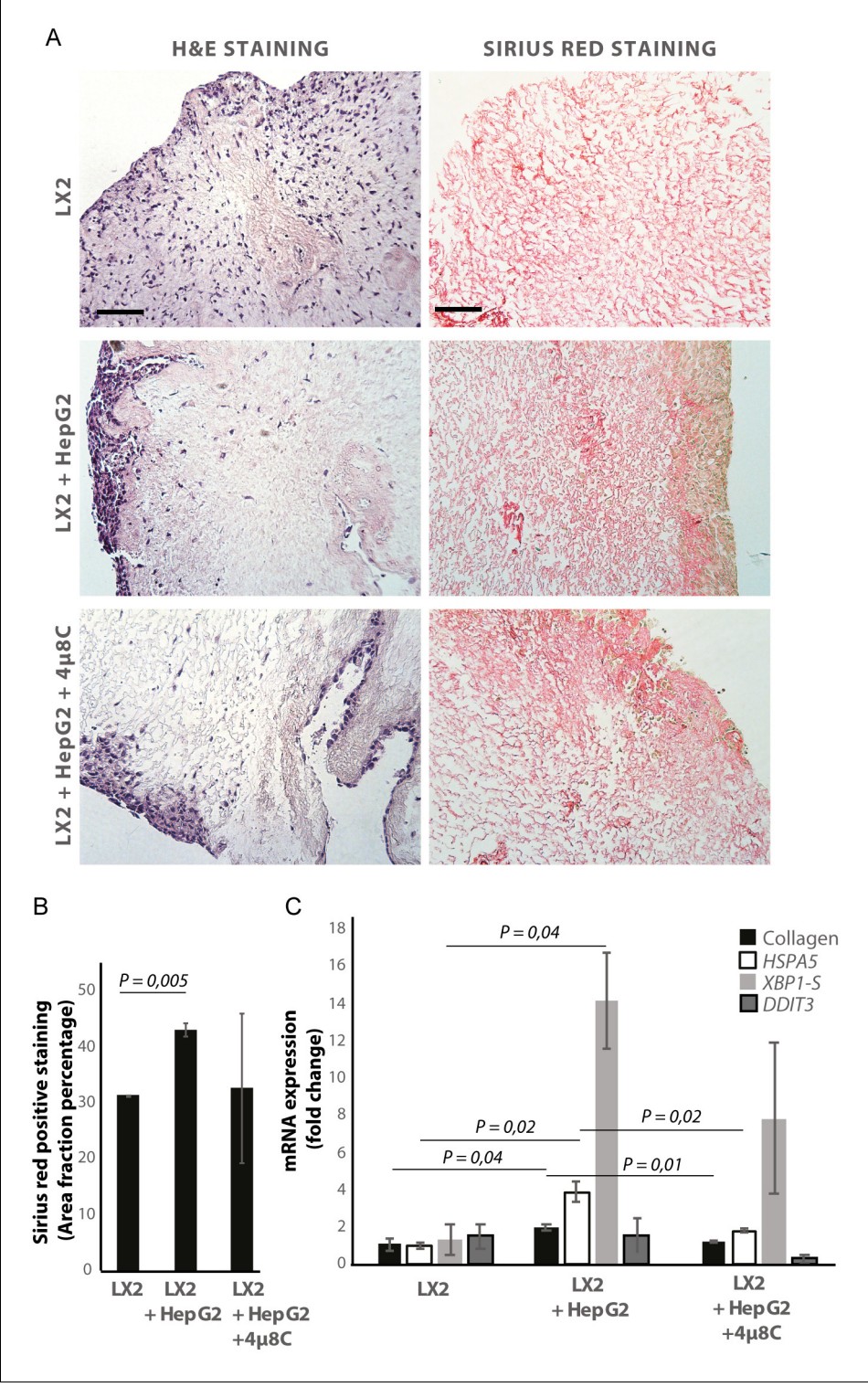

**Figure 5.** Inhibiting IRE1α decreases stellate cell activation in human liver 3D scaffolds engrafted with stellate cells and tumor cells. (**A**) Representative images of H and E and Sirius red stained slides of decellularized human liver scaffolds engrafted with LX2 stellate cells and HepG2-tumor cells treated with 4μ8C or control. (**B**) Quantification of collagen-stained area fraction of liver scaffolds engrafted with LX2 stellate cells and HepG2-tumor cells treated with 4μ8C or control. (**C**) mRNA-expression of the stellate cells activation marker collagen and ER-stress markers HSPA5, spliced *XBP-1* (XBP1-S), and *DDIT3* in liver scaffolds engrafted with stellate cells (LX2) and cancer cells

*Figure 5 continued on next page*

*Figure 5 continued*

(HepG2), treated with 4μ8C or control. p-Values were calculated via ANOVA from three biological replicates per group, scale bars = 100 μm.

combination with LX2 stellate cells (*Figure 6E*), there was no difference in proliferation between spheroid-monocultures and spheroid-co-cultures in the Huh7-cells (*Figure 6F*). Treatment with 4μ8C significantly decreased proliferation of the tumor spheroids consisting of tumor cells (Huh7 or HepG2) and stellate cells (LX2), while tumor spheroid monocultures were not affected by 4μ8C. Similarly, *PCNA*-mRNA-expression significantly increased in human liver scaffolds engrafted with HepG2 and LX2-cells, compared to those engrafted with only tumor cells (*Figure 7A*). Treatment with 4μ8C significantly decreased *PCNA*-mRNA-expression in the LX2+HepG2 liver scaffolds, whilst not affecting those engrafted with only tumor cells. This further confirms our hypothesis that 4μ8C can affect tumor cell proliferation indirectly, namely by blocking the activation of stellate cells and thus impairing the interaction between tumor and stromal cells.

We measured the mRNA-expression of hepatocyte-nuclear-factor-4-alpha (*HNF4A*), which is a liver function marker that is correlated to a favorable outcome for HCC-patients (*Hang et al., 2017*). While co-engraftment of LX2 and HepG2-cells in the liver scaffolds only lead to a marginal increase of *HNF4A*, treatment with 4μ8C significantly increased *HNF4A*-mRNA-expression, thus suggesting an overall improvement of liver function and possibly improved prognosis (*Figure 7B*). Immunohistochemical staining of EPCAM and KI67, showed that the HCC-cells have successfully engrafted the entire surface of the scaffolds and that 4μ8C decreased proliferation (*Figure 7C*).

## Pharmacological inhibition of IRE1α decreases tumor cell migration in stellate cell – tumor cell co-cultures

Co-culturing HepG2 and Huh7-tumor cells with LX2-cells in the transwell assays significantly increased mRNA-expression of the pro-metastatic marker *MMP9* in HepG2-cells (*Figure 8A*) and *MMP1* in HepG2 and Huh7-cells (*Figure 8B*). Adding 4μ8C significantly decreased the mRNA-expression of *MMP1* in HepG2+LX2 and Huh7+LX2 transwell co-cultures, while a non-significant decrease of *MMP9* mRNA-expression was seen in Huh7+LX2 transwell co-cultures. To assess whether this reduction in mRNA-expression of pro-metastatic markers has a functional effect on cell migration, a scratch wound assay was performed on confluent layers of mono-cultures (HepG2 or LX2) or tumor cell (HepG2) – stellate cell (LX2) co-cultures (*Figure 8C*). To visualize closing of the scratch wound by each individual cell type, cells were fluorescently labeled using CellTracker Green (tumor cells) or CellTracker Red (LX2 stellate cells) (*Figure 8D*). Tumor-stellate cell co-cultures were the most efficient to close the scratch wound (*Figure 8E*). This was significantly inhibited when co-cultures were treated with 4μ8C. We also observed a direct effect of 4μ8C on LX2 and HepG2-migration, since treatment with 4μ8C lead to a significant reduction in wound closure after 24 hr, compared to untreated controls. It is important to note that traditional scratch wound assays cannot distinguish between proliferation and migration (*Cormier et al., 2015*). To overcome this limitation (*Bise et al., 2011*), we counted the individual number of cells in the middle of the wound area (*Figure 8F and G*). No significant difference was seen between HepG2 or LX2-cells within the wound area of HepG2-LX2 co-cultures after 24 hr (*Figure 8F*). However, 4μ8C-treatment significantly decreased migration of HepG2-cells and LX2-cells inside the scratch wound in co-cultures, while not affecting mono-cultures (*Figure 8G*).

Metastasis is usually a result of directed migration and chemotaxis toward physical and biochemical gradients within the tumor stroma (*Oudin and Weaver, 2016*). We used a microfluidic-based device for studying cell migration toward a stable gradient of chemotactic factors, such as FBS. 4μ8C significantly decreased total migration (*Figure 8—figure supplement 1A–C*) and directional migration towards FBS (*Figure 8—figure supplement 1B and D*) of HepG2-cells co-cultured with LX2-cells. Similarly, inhibition of IRE1α with 4μ8C significantly decreased total migration (*Figure 8—figure supplement 1E and G*) and directional migration toward FBS (*Figure 8—figure supplement 1F and H*) of LX2-cells co-cultured with HepG2-cells. Overall, these data suggest that stellate cells increase proliferation and pro-metastatic potential of tumor cells and blocking the IRE1α-RNase activity decreases tumor cell proliferation and migration.

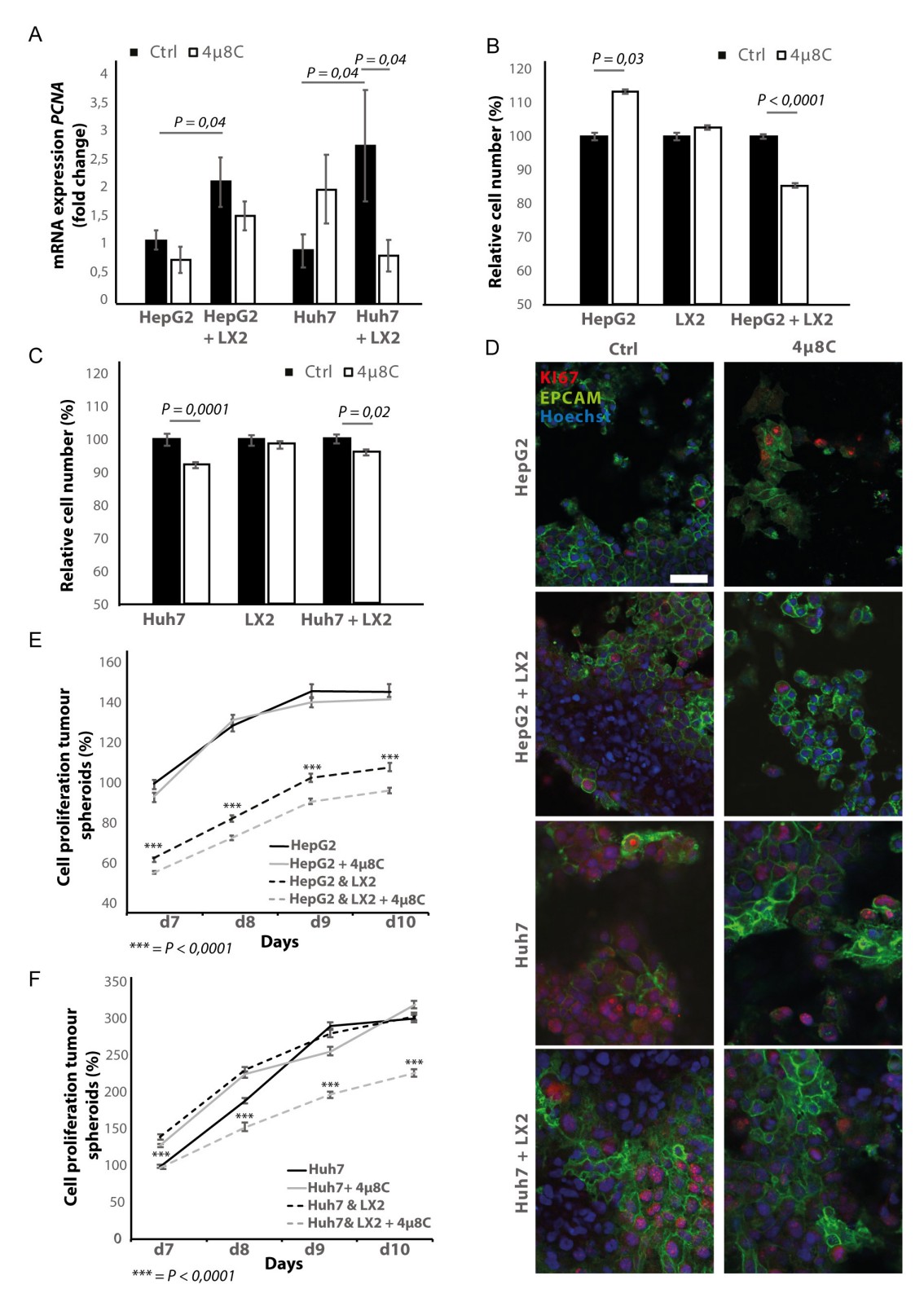

**Figure 6.** Inhibition of IRE1α decreases tumor cell proliferation. (**A**) PCNA mRNA-expression of HepG2 or Huh7-cells grown with LX2-cells in transwell inserts and treated with the IRE1α-inhibitor 4µ8C or control. (**B**) Relative cell number of LX2 and HepG2 or (**C**) LX2 and Huh7-cells treated with 4µ8C or control. (**D**) Representative images of tumor cells (HepG2 or Huh7) and LX2-stellate cells stained with antibodies against the HCC-marker EPCAM and

*Figure 6 continued on next page*

*Figure 6 continued*

the proliferation marker KI67. (E) Cell proliferation of HepG2 or HepG2+LX2 spheroids and (F) Huh7 or Huh7+LX2 spheroids treated with 4µ8C or control. p-Values were calculated via the Student's T-test from nine biological replicates per group, scale bars = 50 µm.

## Silencing of IRE1α in stellate cells decreases tumor cell proliferation and migration in co-cultures

To investigate whether the effect of blocking IRE1α is due to a direct effect on the tumor cells or because of an indirect effect via stellate cells, we transfected the stellate-line LX2 and the tumor cell lines Huh7 and HepG2 with an-siRNA targeting IRE1α, prior to co-culturing. In the LX2-cells, transfection efficiency was determined via qPCR and showed a 50% reduction in the ERN1-mRNA-expression (*Figure 9A*) compared to mock-transfected (Scr) controls. In the transwell co-culturing assay, we found that silencing IRE1α in the LX2-cells significantly decreased *PCNA*-mRNA-expression in HepG2-cells (*Figure 9B*). Silencing IRE1α in the LX2-cells also lead to a significant reduction of proliferation in LX2-HepG2 co-cultures (*Figure 9C*) and LX2-HepG2 spheroids (*Figure 9—figure supplement 1A*). Immunocytochemical staining with αSMA-antibodies (*Figure 9—figure supplement 1B*), confirmed a significant reduction of αSMA after si-IRE1α-transfection of LX2-stellate cells in HepG2-LX2 spheroid co-cultures (*Figure 9—figure supplement 1C*). A scratch wound assay on HepG2-LX2 co-cultures verified that silencing of IRE1α in LX2-cells significantly reduced wound closure compared to non-transfected and mock-transfected stellate cells (*Figure 9—figure supplement 1D–E*). Overall, these data confirm that blocking the IRE1α-pathway in hepatic stellate cells decreases proliferation and pro-metastatic potential of tumor cells in co-cultures.

In the cancer-cells, transfection efficiency was determined via qPCR and showed a 40% reduction in the *ERN1*-mRNA-expression in HepG2-cells and 65% in the Huh7-cells (*Figure 9D*). Interestingly, we found that silencing IRE1α in the HepG2-cells led to a significant reduction of proliferation in LX2-HepG2 co-cultures and in the HepG2-monocultures, while silencing IRE1α in the Huh7-cells led to a significant increase in both mono- and co-cultures (*Figure 9E*). These results indicate that silencing IRE1α in the tumor cells also directly affects tumor cell proliferation, but the effect seems to be cell line dependent.

## Inhibiting IRE1α affects the generation of reactive oxygen species

To study if the observed effects of inhibiting IRE1α are through an effect on the generation of reactive oxygen species (ROS), we measured intracellular ROS-levels in 4µ8C-treated (*Figure 10A*) and *IRE1α*-silenced cell lines (*Figure 10B*). Treatment with 50 µM 4µ8C and 100 µM 4µ8C significantly decreased intracellular ROS-levels in LX2, HepG2 and Huh7-cells (*Figure 10A*). No differences were observed between the two concentrations (*Figure 10A*). In the si-IRE1α transfected cells, the effect on ROS-generation seemed to be dependent on the cell type (*Figure 10B*). Transfecting LX2-cells with si-IRE1α led to a significant decrease in intracellular ROS, while this caused a significant increase in the HepG2-cell line (*Figure 10B*). No significant differences were seen in the Huh7-cells (*Figure 10B*). Treatment with 4µ8C further decreased ROS-generation in all transfected cell lines (*Figure 10B*).

## Discussion

There is increasing evidence that ER-stress and activation of the UPR play an essential role during hepatic inflammation and chronic liver disease. We have previously shown that inhibition of IRE1α prevents stellate cell activation and reduces liver cirrhosis in vivo (*Heindryckx et al., 2016*). In this report, we further define a role of the IRE1α-branch of the UPR in the interaction between tumor cells and hepatic stellate cells. We also show that IRE1α could form a valuable therapeutic target to slow down the progression of hepatocellular carcinoma, both through the effect on stromal cells and via the direct effect on cancer cells.

Activated stellate cells play an important role in promoting tumorigenesis and tumors are known to secrete cytokines, such as TGFβ, which activate stellate cells and thereby creates an environment that helps to sustain tumor growth (*Heindryckx, 2014*). Since over 80% of HCC arises in a setting of chronic inflammation associated with liver fibrosis, targeting the fibrotic tumor micro-environment is

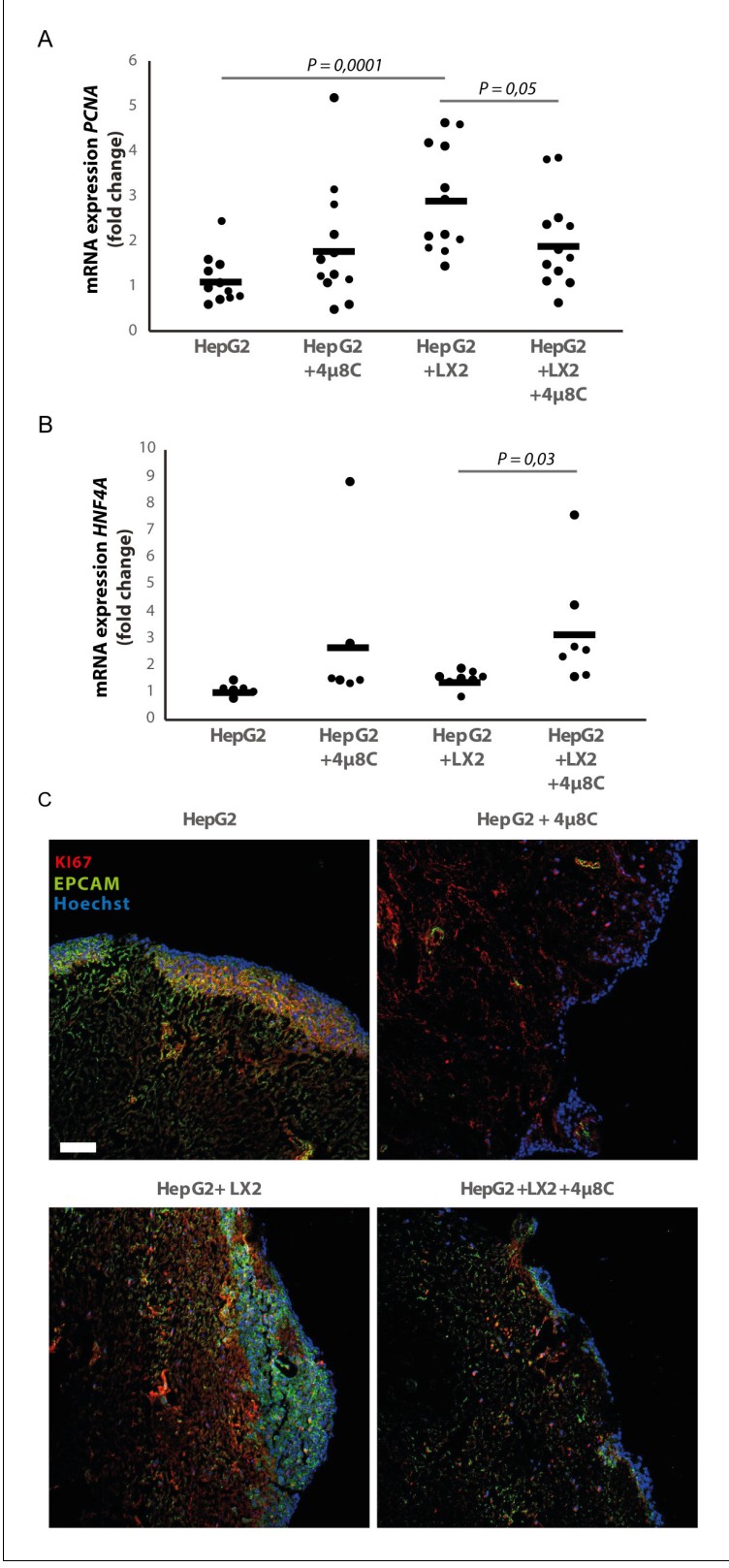

**Figure 7.** Inhibition of IRE1α decreases cell proliferation and improves liver function in human liver scaffolds engrafted with stellate cells and tumor cells. (**A**) PCNA and (**B**) HNF4A expression of human liver scaffolds engrafted with HepG2-tumor cells and LX2-stellate cells, treated with 4µ8C or control. (**C**) Representative images of tumor cells (HepG2) and LX2-stellate cells stained with antibodies against the HCC-marker EPCAM and the

*Figure 7 continued*

proliferation marker KI67. p-Values were calculated via ANOVA on three biological replicates per group, scale bars = 100 µm.

often proposed as a valuable therapeutic strategy for HCC-patients (*Coulouarn and Clément, 2014*). We and others have shown that ER-stress plays an important role in stellate cell activation and contributes to the progression of liver fibrosis (*Heindryckx et al., 2016*; *Koo et al., 2016*; *Hernández-Gea et al., 2013*; *Kim et al., 2016*; *Mao and Fan, 2015*). The mechanisms by which the UPR promotes stellate cell activation have been attributed to regulating the expression of c-MYB (*Heindryckx et al., 2016*), increasing the expression of SMAD-proteins (*Koo et al., 2016*) and/or by triggering autophagy (*Kim et al., 2016*; *Mao and Fan, 2015*).

In our study, we show that IRE1α plays an important role in stellate cell – tumor cell interactions and that pharmacological inhibition of IRE1α-endoribonuclease activity slows down the progression of HCC in vivo. We demonstrate that tumor cells can induce the IRE1α-branch of the UPR in hepatic stellate cells, thereby contributing to their activation and creating an environment that is supportive for tumor growth and metastasis. By co-culturing stellate cells with tumor cells, we mainly observed an increase of the IRE1α-branch of the UPR; however, it is important to note that HepG2-cells also significantly induced mRNA-expression of *EIF2AK3*, while Huh7-cells seemed to induce *DDIT3* in the LX2-cells. These results indicate that ATF6α and PERK-pathways may also play an important role in the interaction between stellate cells and tumor cells. In our study, we also observe that overall levels of XBP1 (spliced and unspliced) were very low in the LX2 monocultures and LX2 + HepG2 co-cultures treated with 4µ8C. This is likely the result of low baseline levels of total XBP1 in these conditions. Several studies have shown that constitutive levels of total XBP1 can be low (*Zeng et al., 2009*) and that the levels of spliced and unspliced XBP1 can both increase during ER-stress (*Yoshida et al., 2001*; *Cassimeris et al., 2019*; *Kishino et al., 2017*). The conditions where we observe low levels of both unspliced and spliced XBP1 correspond to those where we expect to see low levels of IRE1α activation and thus possibly suggest that ER-stress increased the levels of total XBP1 in hepatic stellate cells. Another unexpected finding in our study is the predominant cytoplasmic localization of spliced XBP1 in liver tissue. Spliced XBP1 contains a nuclear localization signal and a transcriptional activation domain, which can activate the transcription of the UPR target genes. In our study, we do not observe a clear nuclear expression of spliced XBP1, which is in contrast to the study of *Yoshida et al., 2006*, which shows that spliced XBP1 predominantly localizes in the nucleus of HeLa-cells exposed to acute ER-stress. This study also describes a mechanism whereby unspliced-XBP1 forms a complex with the spliced isoform, thereby exporting it from the nucleus to the cytoplasm, resulting in subsequent degradation by the proteasome. However, this event has been described during the recovery phase of an acute ER-stress event. In our mouse model, we treated mice with a hepatocarcinogenic compound for 25 weeks, resulting in a chronic inflammation and a subsequent activation of IRE1α-dependent ER-stress pathways. It is therefore not unlikely that different cells in this model are experiencing different phases of ER-stress and recovery. At a higher magnification, it becomes clear that the expression of spliced XBP1 is not only cytoplasmic but some staining appears peri-nuclear and nuclear. This could represent different stages of ER-stress activation and recovery in different cell populations; however, more experiments would be needed to verify this hypothesis.

Our results show that TGFβ-secretion by tumor cells could be in part responsible for activating stellate cells and for inducing the IRE1α-branch of the UPR. However, this seems to depend on the cell lines used, as the effect was not seen in the LX2 and HepG2 co-cultures. In these co-cultures, an autocrine signaling mechanism may be playing a role in the LX2-cells and the HepG2 cells may even prevent this. One possible alternative mechanism is through CCN protein upregulation, as this has been shown to induce ER-stress and UPR-activation in both stellate cells and hepatocytes by in vitro and in vivo approaches. CCN proteins are ECM-associated secreted proteins which play a role in a with a wide array of important functions, such as wound healing and tumorigenesis (*Park et al., 2019*). Adenoviral CCN gene transfer and overexpression of CCN proteins have been shown to induce ER-stress-mediated stellate cell senescence and apoptosis in later stages of fibrosis, consequently contributing to fibrosis resolution (*Borkham-Kamphorst et al., 2016*; *Borkham-Kamphorst et al., 2018*). While ER stress is known to play a key role in stellate cell activation and

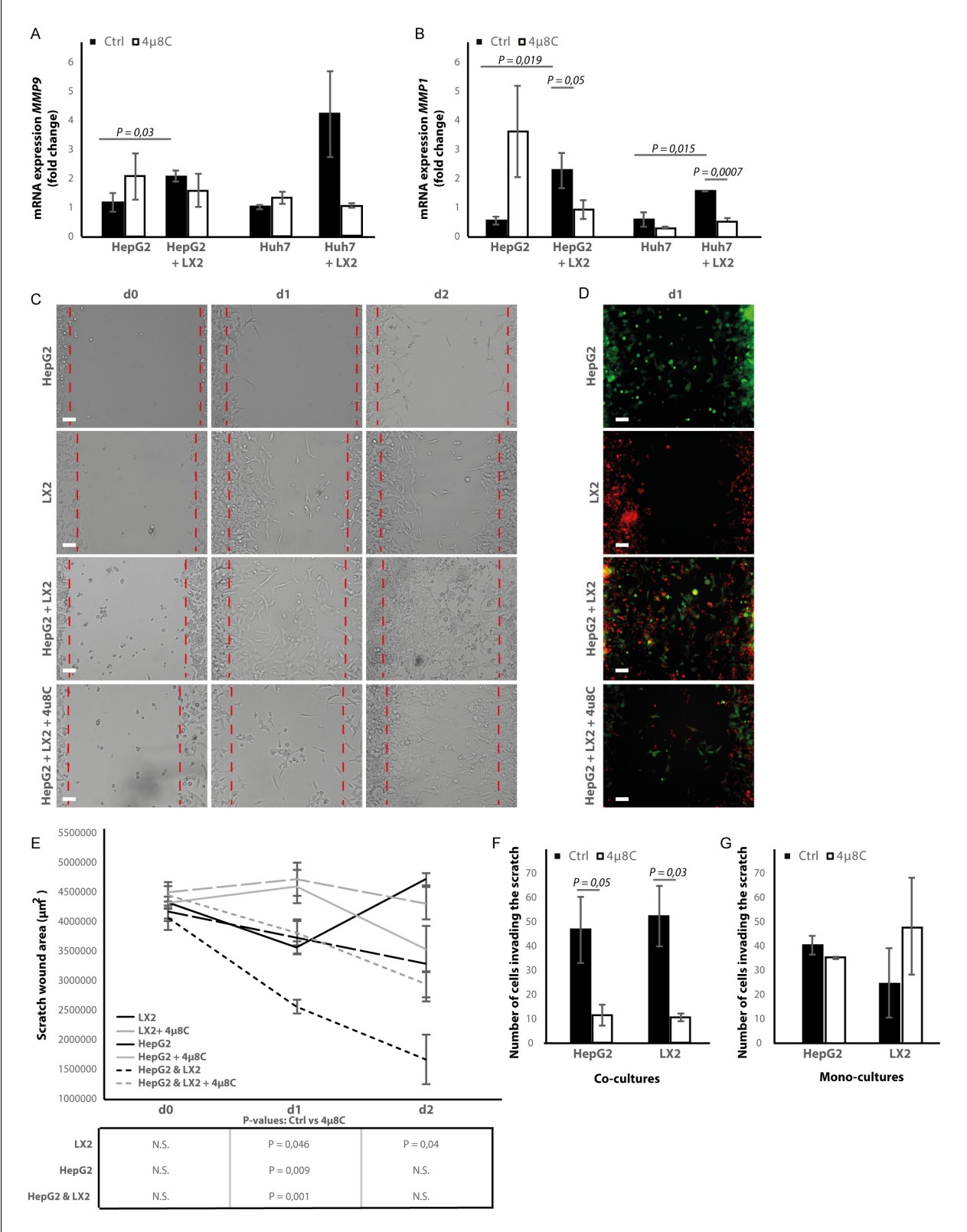

**Figure 8.** Inhibition of IRE1α decreases cell migration. (**A**) mRNA-expression of pro-metastatic markers MMP9 and (**B**) MMP1 in HepG2 and Huh7-cells co-cultured with LX2-cells and treated with 4μ8C or control. (**C**) Scratch wound on HepG2-cells and LX2-cells treated with 4μ8C or control. (**D**) Images of Cell Tracker stained HepG2-cells (Green) and LX2-cells (Red) invading the scratch area. (**E**) Quantification of wound size in HepG2-cells and LX2-cells treated with 4μ8C or control. (**F**) Number of HepG2-cells and LX2-cells invading the scratch wound after 24 hr in co-cultures and (**G**) mono-cultures.

*Figure 8 continued on next page*

*Figure 8 continued*

p-Values were calculated via the Student's T-test from 10 biological replicates per group (panel A and B) or six biological replicates per group (panel E-G), scale bars = 120 μm.

The online version of this article includes the following figure supplement(s) for figure 8:

**Figure supplement 1.** Inhibiting IRE1α decreases chemotaxis.

hepatocyte apoptosis during the fibrosis progression, inducing ER-stress-mediated apoptosis in activated stellate cells in advanced stages of fibrosis could be a relevant therapeutic strategy to attenuate liver fibrosis (*Borkham-Kamphorst et al., 2016*; *Borkham-Kamphorst et al., 2018*).

Activated stellate cells are known to enhance migration and proliferation of tumor cells in vitro (*Song et al., 2016*) and in vivo (*Amann et al., 2009*), by producing ECM-proteins and by secreting

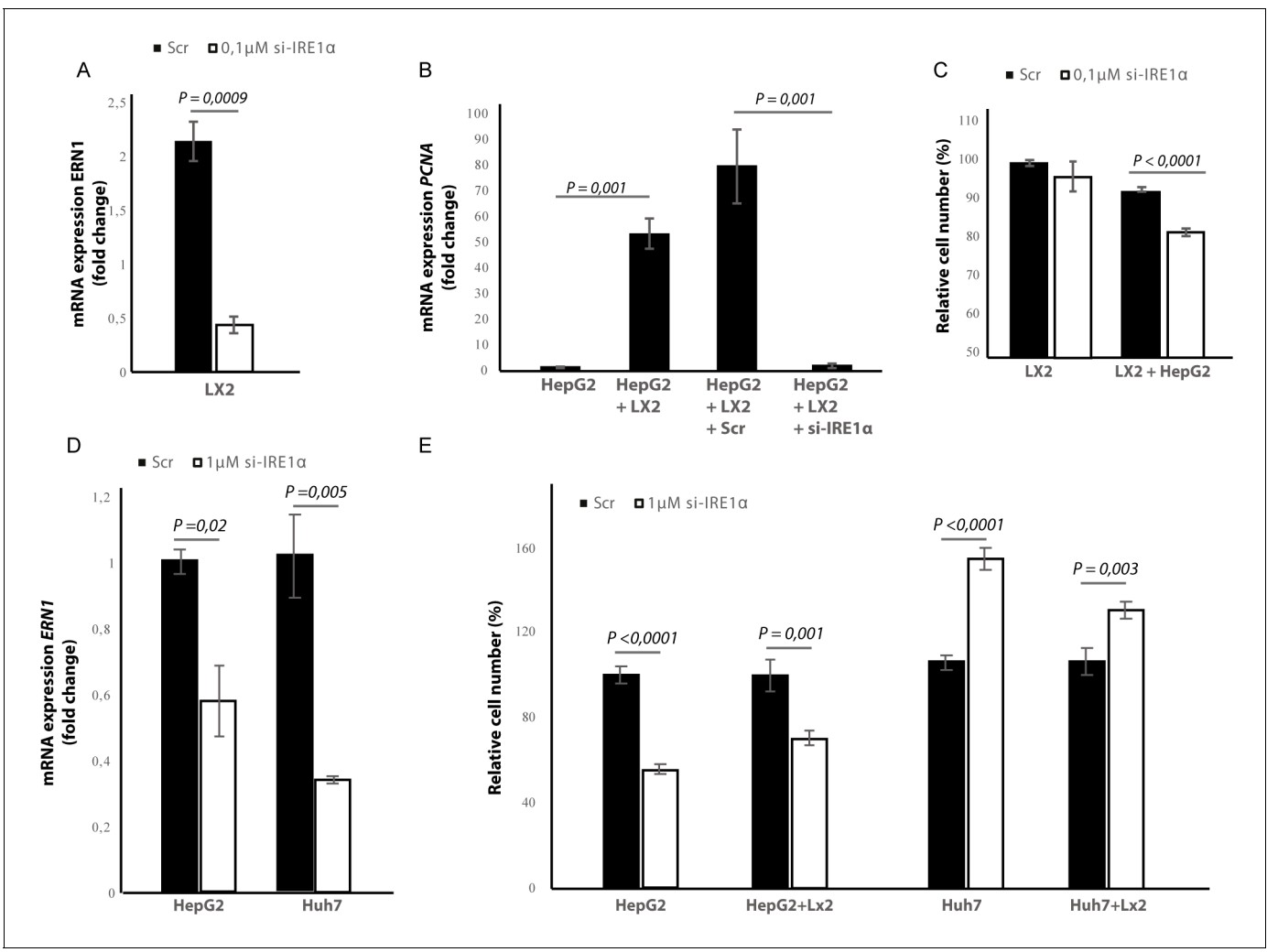

**Figure 9.** Silencing IRE1α in LX2-cells mimics 4μ8C. (A) ERN1-mRNA-expression of LX2-cells transfected with IRE1α-siRNA (si-IRE1α) or mock-transfected (Scr) (B) PCNA-mRNA-expression of HepG2-cells co-cultured with IRE1α-silenced LX2-cells or controls (C). Relative cell numbers in co-cultures of HepG2-cells and IRE1α-silenced LX2-cells or controls. (D) ERN1-mRNA-expression of HepG2- and Huh7-cells transfected with IRE1α-siRNA (si-IRE1α) or mock-transfected (Scr). (E) Relative cell numbers in co-cultures LX2-cells or and si-RNE. Transfected HepG2 or Huh7 cells or mock-transfected controls (Scr). p-Values were calculated via the Student's T-test from three biological replicates per group (panel **A**, **B and D**) or six biological replicates (panel C and E).

The online version of this article includes the following figure supplement(s) for figure 9:

**Figure supplement 1.** Proliferation and migration after silencing IRE1α in LX2-cells.

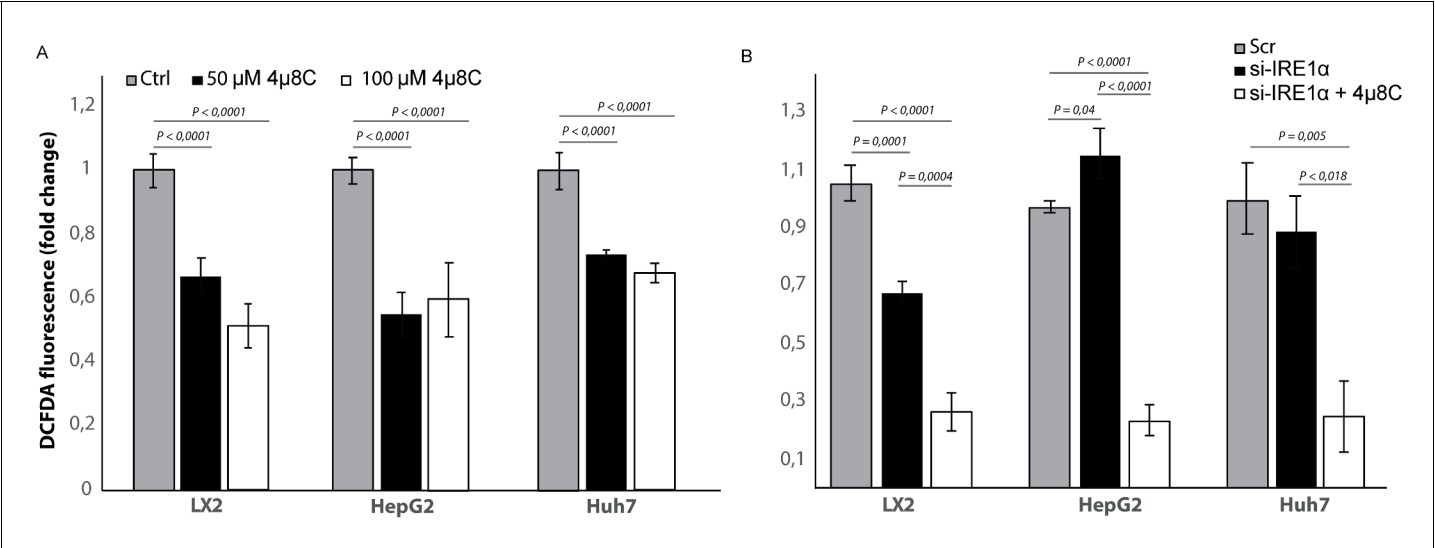

**Figure 10.** Inhibiting IRE1α alters generation of ROS. (**A**) intracellular ROS-levels in LX2, HepG2, and Huh7 cells treated with 50 µM 4µ8C, 100 µM 4µ8C or controls. (**B**) intracellular ROS-levels in LX2, HepG2 and Huh7 cells transfected with IRE1α-siRNA (si-IRE1α) or mock-transfected (Scr). p-Values were calculated via the Student's T-test from three biological replicates per group.

growth factors. Extracellular matrix proteins such as collagen can act as a scaffold for tumor cell migration (*Han et al., 2016*), alter the expression of MMPs (*Song et al., 2016*) and induce epithelial-mesenchymal transition (*Kumar et al., 2014*). Activated stellate cells are also an important source of hepatocyte growth factor, which promotes proliferation, cell invasion, and epithelial-mesenchymal transition via the c-MET signaling pathway (*Liu et al., 2016*). Interestingly, blocking IRE1α in the stellate cell population reduced tumor-induced activation toward myofibroblasts, which then decreases proliferation and migration of tumor-cells in co-cultures. This suggests that targeting the microenvironment using an ER-stress inhibitor could be a promising strategy for patients with HCC.

The UPR has been described as an essential hallmark of HCC (*Shuda et al., 2003*), although its role within tumorigenesis remains controversial (*Vandewynckel et al., 2013*). While a mild-to-moderate level of ER-stress leads to activation of the UPR and enables cancer cells to survive and adapt to adverse environmental conditions, the occurrence of severe or sustained ER-stress leads to apoptosis. Both ER-stress inhibitors and ER-stress inducers have therefore been shown to act as potential anti-cancer therapies (*Corazzari et al., 2017*). A recent study by *Wu et al., 2018*, demonstrated that IRE1α promotes progression of HCC and that hepatocyte-specific ablation of IRE1α results in a decreased tumorigenesis. In contrast to their study, we found a greater upregulation of actors of the IRE1α-branch within the stroma than in the tumor itself and identified that expression of these IRE1α-markers was mainly localized within the stellate cell population. An important difference between both studies is the mouse model that was used. While Wu et al used a single injection of DEN, we performed weekly injections, causing tumors to occur in a background of fibrosis, similar to what is seen in patients (*Heindryckx et al., 2010*). Our in vitro studies with mono-cultures confirm that 4µ8C and transfection with si-IRE1α also has a direct effect on proliferation, migration, and intracellular levels of ROS in HCC-cells – similar to the findings of Wu et al - and the response seems to depend on the tumor cell line. Adding 4µ8C to HepG2-cells significantly increased proliferation, while a significant decrease was seen in the Huh7-cells. This difference in response could be due IRE1α's function as a key cell fate regulator. On the one hand, IRE1α can induce mechanisms that restore protein homeostasis and promote cytoprotection, whereas on the other hand IRE1α also activates apoptotic signaling pathways. How and when IRE1α exerts its cytoprotective or its pro-apoptotic function remains largely unknown. The duration and severity of ER-stress seems to be a major contributor to the switch toward apoptosis, possibly by inducing changes in the conformational structure of IRE1α (*Ghosh et al., 2014*). The threshold at which cells experience a severe and prolonged ER-stress that would induce apoptosis could differ between different cell lines, depending on the translational capacity of the cells (e.g. ER-size, number of chaperones and the amount of

degradation machinery) and the intrinsic sources that cause ER-stress (*Cubillos-Ruiz et al., 2017*). A study of *Li et al., 2012*, has specifically looked at how IRE1α regulates cell growth and apoptosis in HepG2-cells. Similar to our findings, they discovered that inhibiting IRE1α enhances cell proliferation, while over-expression of IRE1α increases the expression of polo-like kinase, which leads to apoptosis. Interestingly, polo-like kinases have divergent roles on HCC-cell growth depending on which cell line is used, which could explain the different response to 4μ8C in Huh7 and HepG2-cells (*Pellegrino et al., 2010*). Studies on glioma cells show that IRE1α regulates invasion through MMPs (*Auf et al., 2010*). In line with these results, we also detected a reduction of *MMP1*-mRNA expression after 4μ8C-treatment and observed a direct effect on wound closure in HepG2-cells. These results indicate that IRE1α could play a direct role in regulating tumor cell invasion, in addition to its indirect effect via stellate cells. This is also in line with our findings that silencing IRE1α in the tumor cells affects tumor cell proliferation, although this effect seems to be cell line dependent.

Another possible mechanism that explains the cell line specific differences in response to inhibiting IRE1α, is through the generation of ROS. Studies have shown that IRE1α plays an important role in mediating ROS-generation (*Abuaita et al., 2015*) and 4μ8C has been described as a potent ROS-scavenger (*Chan et al., 2018*). IRE1α generates ROS through $Ca^{2+}$-mediated signaling between the IRE1α-InsP3R pathway in the ER and the redox-dependent apoptotic pathway in the mitochondrion, as well as via activation of CHOP, BIP and through XBP1-splicing (*Riaz et al., 2020*; *Zeeshan et al., 2016*). In line with these findings, we found a significant reduction in intracellular ROS-levels after treatment with 4μ8C in LX2, HepG2 and Huh7-cells. Interestingly, a similar reduction in ROS-generation as in the 4μ8C-treated LX2-cells was seen after transfection of LX2-cells, while an increase of ROS-generation was noted in the transfected HepG2-cells. These results indicate that the reduction in ROS could partially be explained through the decreased activation of the IRE1α-pathway in the LX2-cells. However, how the IRE1α-pathway affects the generation of ROS, seems to be cell-type dependent, as we see different results in the different cell lines we tested. This is in line with previous studies, which also observed this cell line dependent effect on IRE1α-dependent ROS-generation (*Chan et al., 2018*; *Zeeshan et al., 2016*). The HepG2 and Huh7-cell lines used in this study are known to have different sensitivities to doxorubicin, a property that has been ascribed to their differences in intracellular ROS-generation after treatment with this chemotherapeutic agent (*Dubbelboer et al., 2019*). Alterations in oxidative stress can affect cell proliferation, specifically in cancer cells and stellate cells (*Montiel-Duarte et al., 2004*). In addition, ROS is one of the critical mediators of stellate cell activation and ECM-production (*Kong et al., 2019*). Oxidative stress has been recognized as one of the key factors in the pathogenesis of HCC and treatment strategies aiming at controlling oxidative stress have shown promising pre-clinical results (*Takaki and Yamamoto, 2015*; *Uchida et al., 2020*). Therefore, an IRE1α-mediated regulation of ROS-generation might be a contributing factor that explains our findings on stellate cell activation and tumor cell proliferation after inhibiting IRE1α with 4μ8C or transfection. However, more research is necessary to further elucidate the role of IRE1α in mediating ROS-generation in different cell types. In addition, since we see a potent decrease on ROS-levels after treatment with 4μ8C, even in the cells that were transfected with si- IRE1α, we cannot exclude that – at least part – of our results could be explained through the off-target effect of 4μ8C as a ROS-scavenger. Inhibiting oxidative stress has been shown to attenuate tumor progression in different pre-clinical models for HCC and ROS is a known contributor to the chronic liver disease and HCC (*Tien Kuo and Savaraj, 2006*; *Klieser et al., 2019*). Further research is necessary to assess to which extent the ROS-scavenging effect in our study has influenced cancer progression.

In conclusion, the aim of this study was to define the role of IRE1α in the cross-talk between hepatic stellate cells and tumor cells in liver cancer. We show that pharmacologic inhibition of the IRE1α-signaling pathway decreases tumor burden in a DEN-induced mouse model for HCC. Using several in vitro 2D and 3D co-culturing methods, we show that tumor cells can induce the IRE1α-branch of the ER-stress pathways in hepatic stellate cells and that this contributes to their activation. Blocking IRE1α-in these hepatic stellate cells prevents their activation. This then contributes to a decreased proliferation and migration of tumor cells in co-cultures, in addition to the direct effect of inhibiting IRE1α in tumor cells. Our results indicate that there are cell-line-specific differences in the response to IRE1α-inhibition, including intercellular variations in how blocking IRE1α affects the generation of ROS.

# Materials and methods

### Key resources table

| Reagent type (species) or resource | Designation | Source or reference | Identifiers | Additional information |
|---|---|---|---|---|
| Strain, strain background (*Mus musculus*) | Sv129 mice | Taconic | 129S6 | HCC mouse model, *Heindryckx et al., 2010*; *Heindryckx et al., 2012* |
| Cell line (*Homo sapiens*) | HepG2 | ATCC | HB-8065 | |
| Cell line (*Homo sapiens*) | Huh7 | Gifted, Karolinska institute | | |
| Cell line (*Homo sapiens*) | LX2 | Sigma-Aldrich | SCC064 | |
| Transfected construct (human) | si-IRE1α | ThermoFisher | s200432 | 0,1–1 µM |
| Transfected construct (human) | Si-Ctrl; Scr | ThermoFisher | 4390843 | 0,1–1 µM |
| Antibody | KI67 (rat monoclonal) | eBioscience | SolA15 | 1:100 |
| Antibody | EPCAM (rabbit polyclonal) | Abcam | ab71916 | 1:100 |
| Antibody | Spliced XBP1 (goat monoclonal) | Abcam | Ab85546 | 1:50 |
| Antibody | Total XBP1 (Rabbit polyclonal) | Abcam | Ab37152 | 5 µg/ml |
| Antibody | IRE1a (rabbit polyclonal) | Abcam | Ab37073 | 1 µg/ml |
| Antibody | p-IRE1 (rabbit polyclonal) | AbNova | PAB12435 | 1:100 |
| Antibody | αSMA (Rabbit Polyclonal) | ThermoFisher | 710487 | 1:200 |
| Antibody | αSMA (Goat monocolonal) | Abcam | Ab21027 | 1–2 µg/ml |
| Antibody | BIP (goat polyclonal) | Abcam | Ab21027 | 1 µg/ml |
| Antibody | Vinculin (Mouse monoclonal) | ThermoFisher | 14-9777-82 | 1–5 µg/ml |
| Peptide, recombinant protein | *Pst*-I | ThermoFisher | ER0615 | |
| Commercial assay or kit | Pierce BCA- protein assay kit | ThermoFisher | 233225 | |
| Commercial assay or kit | EZNA RNA isolation Kit II | VWR | R6934-02 | |
| Commercial assay or kit | RNeasy Universal Mini Kit | Qiagen | 73404 | |
| Commercial assay or kit | Diva Decloacker solution | Biocare | DV2004 | |
| Commercial assay or kit | DCFDA - Cellular ROS Detection Assay Kit | Abcam | ab113851 | |
| Chemical compound, drug | N-Nitrosodiethylamine, DEN | Sigma-Aldrich | 1002877809 | |
| Chemical compound, drug | 4µ8C | Sigma-Aldrich | SML0949-25MG | *Heindryckx et al., 2016* |
| Chemical compound, drug | SB-431541, TGF-ß receptor inhibitor | Tocris | 1614 | 10 µM |

*Continued on next page*

*Continued*

| Reagent type (species) or resource | Designation | Source or reference | Identifiers | Additional information |
|---|---|---|---|---|
| Chemical compound, drug | Resazurin | Sigma-Aldrich | R7017-1G | 1:80 dilution |
| Commercial assay or kit | Ingenio electroporation solution | Mirus Bio LLC | MIR50114 | Ice-cold |
| Commercial assay or kit | CellTracker Red CMTPX | ThermoFisher | C34552 | 1 µM |
| Commercial assay or kit | CellTracker Green CMFDA | ThermoFisher | C2925 | 1 µM |
| Other | 12-well Corning Costar Transwell plates | Sigma-Aldrich | 3460 | *Calitz et al., 2020* |
| Other | Corning Costar Ultra-Low attachment 96-well plates | Sigma-Aldrich | CLS3471 | *Calitz et al., 2019* |
| Other | CellDirector | GradienTech | 11-001-10 | *Fuchs et al., 2020* |

## Mouse model

A chemically induced mouse model for HCC was used, as previously described (*Heindryckx et al., 2010*; *Heindryckx et al., 2012*). Briefly, 5-week-old male Sv129 mice received intraperitoneal injections once per week with 35 mg/kg bodyweight N-Nitrosodiethylamine (DEN) (1002877809, Sigma-Aldrich, Darmstadt, Germany) diluted in saline. From week 10, mice were injected twice per week with 10 µg/g bodyweight 4µ8C (SML0949-25MG, Sigma-Aldrich, Darmstadt, Germany) in saline. After 25 weeks, mice were euthanized and samples were taken for analysis. The methods were approved by the Uppsala ethical committee for animal experimentation (C95/14). Each group contained eight mice, which generates enough power to pick up statistically significant differences between treatments, as determined from previous experience (*Heindryckx et al., 2010*; *Heindryckx et al., 2012*). Mice were assigned to random groups before treatment.

## Sampling of animal tissue

Liver tissue for mRNA-analysis was divided in non-tumor tissue and tumor tissue, by excising macroscopically visible tumors using surgical micro-scissors. Tissue fragments were then immersed in RNA-later solution (Sigma-Aldrich, Darmstadt, Germany) and incubated on ice for 30 min, followed by snap freezing on dry ice and storage in −80°C. For protein analysis, liver tissue was immediately snap frozen without separating tumor and non-tumor tissue. For paraffin-embedding, half of the left liver lobe was rinsed in ice-cold saline solution and fixed in 4% paraformaldehyde for 24 hr.

## Olink multiplex proximity extension assay

Liver samples were homogenized in ice-cold radioimmunoprecipitation assay (RIPA) buffer (20–188, Merck-Millipore, Solna, Sweden), containing Halt Protease inhibitor cocktail (78425, ThermoFisher Scientific, Stockholm, Sweden). Homogenates were kept on ice for 20–30 min, whilst mixing vigorously to enhance disruption of the cell membranes. The homogenates were then centrifuged (20 min, 13,000 rpm, 4°C) and supernatant containing protein was collected. Supernatant was stored at −20°C until protein measurement. Protein concentration was measured using the Pierce BCA-protein assay kit (233225, ThermoFisher Scientific, Stockholm, Sweden) and all samples were diluted to 1 mg/mL protein in RIPA-buffer. Samples from three biological replicates per group were analyzed with a multiplex proximity extension assay for ninety-two biomarkers in the murine exploratory panel (Olink Bioscience, Uppsala, Sweden) (*Krauthamer et al., 2013*). Samples were loaded at random on the assay plates. Raw data was deposited in Dryad (*Heindryckx, 2020*).

## Cell culture and reagents

The HCC-cell lines (HepG2, ATCC HB-8065 and Huh7, kind gift from Dilruba Ahmed, Karolinska Institute, Sweden) and the human hepatic stellate cell-line LX2 (SCC064, Sigma-Aldrich, Darmstadt, Germany) were cultured at 37°C with 5% $CO_2$ in high glucose Dulbecco modified eagle medium,

GlutaMAX supplemented (DMEM) (31066047, ThermoFisher Scientific, Stockholm, Sweden) supplemented with 1% antibiotic antimycotic solution (A5955-100ML, Sigma-Aldrich, Darmstadt, Germany) followed by 10% and 2% fetal bovine serum (FBS) (10270106, ThermoFisher Scientific, Stockholm, Sweden) for the HCC cell lines and LX2 cell line, respectively (*Calitz et al., 2020*). No FBS was used during starvation and stimulation with growth factors. Misidentification of the three cell lines was checked at the Register of Misidentified Cell Lines, and none of the chosen cell lines were on the list (*Capes-Davis et al., 2010*). Extracted DNA from all our cell lines are sent yearly to Eurofins Genomics (Ebersberg, Germany) for cell line authentication using DNA/STR-profiles. Authentication confirmed the correct identity of each cell line and each cell line was tested negative for mycoplasma contamination.

For transwell co-culturing experiments, cells were grown on 12-well Corning Costar Transwell plates (3460, Sigma-Aldrich, Darmstadt, Germany) with 0.4 µm-pore polyester membrane, allowing the exchange of soluble factors, but preventing direct cell contact (*Calitz et al., 2020*). Cells were detached using 0.05% trypsin-EDTA (15400054, ThermoFisher Scientific, Stockholm, Sweden), re-suspended in growth medium and seeded at a density of $1.0 \times 10^5$ cells per well and $4.0 \times 10^4$ cells per insert. Cells were allowed to attach and left undisturbed for 8 hr, followed by 16 hr of starvation in serum-free medium. Afterwards, fresh starvation medium containing indicated growth factors or substances were added. Cells were exposed for 48 hr to 50 µM or 100 µM 4µ8C or 10 µM SB-431541 (1614, Tocris, Abingdon, UK), as previously described (*Heindryckx et al., 2016*).

3D-tumor spheroids were generated in flat bottom Corning Costar Ultra-Low attachment 96-well plates (CLS3471, Sigma-Aldrich, Darmstadt, Germany) (*Calitz et al., 2019*). After 6 days, spheroids had reached approximately 1 mm$^2$ and 4µ8C was added. Proliferation was monitored during the subsequent 4 days. Tumor spheroids were retrieved from the plates after 10 days and used for immunohistochemical staining.

## Human liver scaffold decellularization and cell culture usage

Human healthy livers were obtained under the UCL Royal Free BioBank Ethical Review Committee (NRES Rec Reference: 11/WA/0077) approval. Informed consent was obtained for each donor and confirmed via the NHSBT ODT organ retrieval pathway (*Mazza et al., 2017*). Liver 3D-scaffolds, were decellularized, sterilized and prepared for cell culture use, as previously described (*Mazza et al., 2017*). LX2 and HepG2-cells, as either mono-cultures or mixed co-culture, were at a seeding density of $2.5 \times 10^5$ cells in volume of 20 µL per scaffold (*Thanapirom et al., 2019*).

## Proliferation

Cell proliferation was monitored *via* a resazurin reduction assay (*Präbst et al., 2017*). Cells were seeded onto Corning 96-well, flat, clear bottom, black plates (CLS3603-48EA, Sigma-Aldrich, Darmstadt, Germany) at a seeding density of $1.0 \times 10^4$ cells for monocultures and a 1:1 ratio of $5.0 \times 10^3$ cells for co-cultures, per well. A 1% resazurin sodium salt solution (R7017-1G, Sigma-Aldrich, Darmstadt, Germany) was added in 1/80 dilution to the cells and incubated for 24 hr, after which fluorescent signal was measured with a 540/35 excitation filter and a 590/20 emission filter on a Fluostar Omega plate reader.

## Transfections

Nucleofection with 0.1–1 µM si-IRE1α (s200432, ThermoFisher Scientific, Stockholm, Sweden), or 0.1 µM siCtrl (4390843, ThermoFisher Scientific, Stockholm, Sweden) was done using Amaxa Nucleofector program S-005 (LX2-cells) or T-028 (HepG2 and Huh7) in ice-cold Ingenio electroporation solution (MIR50114, Mirus Bio LLC, Taastrup, Denmark) on $1.0 \times 10^6$ cells per transfection. Cells were promptly re-suspended in 2 mL DMEM with 10% FBS and left adhere for 6–8 hr, after which the medium was changed to fresh DMEM. Transfection efficiency was checked 24 hr post-transfection by qPCR. Only one si-RNA was used, as this reduced mRNA expression by >40% in all cell lines.

## Migration and chemotaxis

Non-directional migration was assessed using a scratch wound assay, as previously described (*Pinto et al., 2019*). In short, cells fluorescently labeled by using CellTracker dye, according to manufacturer's instructions. Cell pellets were incubated 30 min with 1 µM of CellTracker Red CMTPX

(C34552, ThermoFisher Scientific, Stockholm, Sweden) or 1 µM of CellTracker Green CMFDA (C2925, ThermoFisher Scientific, Stockholm, Sweden). Cells were washed twice in phosphate buffered saline (PBS) (P4417-100TAB, Sigma-Aldrich, Darmstadt, Germany) and seeded in 12-well plates. The cells were left to reach 100% confluency overnight, after which a scratch was created on the confluent cell layer, using a 200 µL pipette tip. Medium was aspirated from each well and replaced by fresh DMEM containing 10% FBS. Invasion of cells into the scratch wound area was monitored using fluorescence microscopy images and light microscopy images. Scratch size was measured by analyzing light microscopy images in ImageJ, using the MRI Wound Healing Tool plug-in (http://dev.mri.cnrs.fr/projects/imagej-macros/wiki/Wound_Healing_Tool). Image analysis was done in ImageJ.

Directional migration was assessed using CellDirector-devices (11-001-10 GradienTech, Uppsala, Sweden), following manufacturer's recommendations (Fuchs et al., 2020). HepG2 and LX2-cells were labeled with CellTracker-dye and left to adhere overnight in the CellDirector-devices. Non-adherent cells were washed away with DMEM and cells were starved for 1 hr prior to commencing experiments. A gradient of 0% to 10% FBS was created with a flow rate of 1.5 µl/min. Cell movement was recorded using an Axiovision 200M microscope (Zeiss, Stockholm, Sweden) for 4 hr and tracked using Axiovision software (Zeiss, Stockholm, Sweden). During the assay cells were kept at 37 °C with 5% $CO_2$.

## Quantitative RT-PCR of mRNA

RNA was isolated from tissue or cell culture using the EZNA RNA isolation Kit II (R6934-02, VWR, Spånga, Sweden) or using QIAzol lysis reagent (79306, Qiagen, Sollentuna, Sweden) and RNeasy Universal Mini Kit (73404, Qiagen, Sollentuna, Sweden) for human liver scaffolds (Mazza et al., 2017). RNA-concentration and purity were evaluated using Nanodrop. Afterwards, 500 ng of mRNA was reverse transcribed using iScript select cDNA synthesis kit (1708897, Bio-rad, Solna, Sweden). Amplifications were done using primers summarized in Supplementary file 1, table 1. mRNA-expression was normalized to 18S, GAPDH and/or TBP1. Fold change was calculated via the delta-delta-CT method, by using the average CT value of three technical replicates.

The procedure to detect the spliced and unspliced isoforms of XBP1 was done by digesting RT-PCR product with the restriction enzyme Pst-I (ER0615, ThermoFisher Scientific, Stockholm, Sweden). This cleaves unspliced-XBP1 containing the Pst-I-cleavage site (CTGCA'G), but leaves the spliced isoform intact. The digestion reaction was stopped after 18 hr by 0,5M EDTA (pH 8.0) and run on a 2.5% agarose (A9539-250G, Sigma-Aldrich, Darmstadt, Germany) gel for 1 hr at 180V. Nucleic acids were visualized by adding GelRed Nucleic Acid Gel Stain (Biotium, Solna, Sweden) in a 1:10,000 dilution to the agarose gels. Agarose gels were scanned using an Odyssey scanner (LI-COR Biotechnology) and bands were quantified using ImageJ.

## Stainings and immunocytochemistry

Tissue samples were fixed in 4% paraformaldehyde for 24 hr and subsequently embedded in paraffin. Cells and tumor spheroids were fixed for 10 min in 4% paraformaldehyde and stored at 4°C until further processing. Paraffin-embedded tissue samples were cut at 5 µm and dried overnight. Sections were de-paraffinized and rehydrated prior to staining. Collagen was stained using the picrosirius red staining with an incubation time of 30 min, followed by 10 min washing in distilled water (Huang et al., 2013). Haematoxylin-eosin (H and E) staining was done according to standard practice (Cardiff et al., 2014). Images were acquired using a Nikon eclipse 90i microscope equipped with a DS-Qi1Mc camera and Nikon plan Apo objectives. NIS-Elements AR 3.2 software was used to save and export images. Quantification of collagen deposition was performed blindly with ImageJ software by conversion to binary images after color de-convolution to separate Sirius Red staining, as previously described (Ruifrok and Johnston, 2001).

Paraformaldehyde fixed cells and spheroids were washed with tris-buffered saline (TBS) (T5030-50TAB, Sigma-Aldrich, Darmstadt, Germany) and blocked for 30 min using 1% bovine serum albumin in TBS + 0,1% Tween 20 (P7949-500ML, Sigma-Aldrich, Darmstadt, Germany). For liver tissue, antigen retrieval was done at 95°C in sodium citrate buffer or Diva Decloacker solution (DV2004, Biocare, Gothenburg, Sweden). Blocking was done using TNB blocking reagent (FP1020, Perkin-Elmer, Hägersten, Sweden) for 45 min and followed by an overnight incubation at 4°C with primary antibodies (Supplementary file 2, table 2). A 40 min incubation was used for the secondary antibody

(Rabbit anti-mouse Alexa Fluor-488 or donkey anti-rabbit Alexa Fluor-633) and cell nuclei were stained with Hoechst for 8 min. Images were taken using an inverted confocal microscope (LSM 700, Zeiss, Stockholm, Sweden) using Plan-Apochromat 20 × objectives and the Zen 2009 software (Zeiss, Stockholm, Sweden). The different channels of immunofluorescent images were merged using ImageJ software. Quantifications were done blindly with ImageJ software by conversion to binary images for each channel and automated detection of staining on thresholded images using a macro.

For histological and immunohistochemical analysis of the human liver scaffolds, 4 µm slides were cut from paraffin embedded blocks. The sections were de-paraffinized and rehydrated prior to staining. To retrieve the antigens, slides were microwaved at high power for 5 min in pre-heated 10 mM sodium citrate buffer, and subsequently left to cool down to room temperature. Following this, a single wash was performed in 100 mM Glycine in PBS, after which the slides were blocked for 2 hr in TNB Blocking Reagent. Slides were then incubated for 2 hr in the following antibodies; KI67 (1:100; SolA15, eBioscience, Stockholm, Sweden), and EPCAM (1:100; ab71916, Abcam, Cambridge, UK). A 1 hr incubation was used for the secondary antibody (goat anti-rat Alexa Fluor 555 and Rabbit anti-mouse Alexa Fluor 488, ThermoFisher, Stockholm, Sweden). Sections were mounted with Fluoromount-G (F4680-25ML, Sigma-Aldrich, Darmstadt, Germany) with DAPI (00-4959-52, Invitrogen, Stockholm, Sweden). Images were taken with using an inverted confocal microscope (LSM 780, Zeiss) using Plan-Apochromat 10 × objectives and the Zen 2009 software (Zeiss).

## Enzyme-linked immune sorbent assay (ELISA)

Medium samples from cells and from the engrafted scaffolds were used to measure TGFβ via ELISA (88-8350-22, ThermoFisher, Stockholm, Sweden), following manufacturer's guidelines. The averages from four biological replicates and two technical replicates were used for calculations.

## SDS-PAGE and western blot

Protein lysates in lysis buffer were mixed with 2x Laemmli buffer and heated to 95 °C for 5 min before being loaded onto a Precast Mini-Protean TGX gels (456–9034, Biorad, Solna, Sweden). After separation, proteins were transferred to an Immobilon-Fl membrane (IPFL0010, Millipore, Solna, Sweden) (*Eaton et al., 2014*). The membrane was blocked using the Intercept (TBS) blocking buffer (927–60001, Li-Cor, Bad Homburg, Germany) diluted 1:4 in PBS, and then incubated with primary and secondary antibodies. After primary and secondary antibody incubation the membrane was washed 3 × 15 min in PBS-T (PBS + 0.1% Tween20). Primary antibodies used were BIP (ab21685, Abcam, Cambridge, UK), XBP1 (ab37152, Abcam, Cambridge, UK), p-IRE1α (PAB12435, Abnova, Heidelberg, Germany) or vinculin (14-9777-82, ThermoFisher, Stockholm, Sweden), diluted in blocking buffer with 0.1% Tween20. Secondary antibodies used were goat-anti-rabbit Alexa 680 (A21088, Invitrogen, Stockholm, Sweden) and goat-anti-mouse IRDye 800 (Rockland, Stockholm, Sweden), diluted 1:20,000 in blocking buffer with 0.1% Tween20% and 0.01% SDS. All incubations were carried out at room temperature for 1 hr or overnight at 4°C. The membranes were scanned using an Odyssey scanner (LI-COR Biotechnology) and band intensities quantified using the Odyssey 2.1 software and normalized to the vinculin signal in each sample (*Eaton et al., 2014*).

## Gene-set enrichment analysis

Gene expression profiles of HCC with a fibrous stroma and without fibrous stroma were accessed through PubMed's Gene Expression Omnibus via accession number GSE31370 (*Seok et al., 2012*). A gene-set containing 79 genes involved in the unfolded protein response was downloaded from The Harmonizome (*Rouillard et al., 2016*) and GSEA software was used to perform a gene-set enrichment assay (*Subramanian et al., 2005*).

## Reactive oxygen species (ROS) assay

Generation of ROS was measured using DCFDA - Cellular ROS Detection Assay Kit (ab113851, Abcam, Cambridge, UK) in a microplate format. Cells were seeded in flat clear bottom black 96-well plates at a density of $1.0 \times 10^5$ cells/well and left to adhere overnight. On the next day, cells were stained with 25 µM DCFDA for 45 min at 37°C, according to manufacturer's guidelines. After 6 hr of treatment, fluorescence was measured at 485 nm excitation and 535 nm emission wavelengths, using

a Fluostar Omega plate reader. Results of the microplate assay are shown as fold change fluorescence from six biological replicates.

## Human protein atlas

Images from biopsies from; HCC patients stained with antibodies against WIPI1 (*The Human Protein Atlas, 2019d*), SHC1 (*The Human Protein Atlas, 2019a*), PPP2R5B (*The Human Protein Atlas, 2019b*) and BIP (*The Human Protein Atlas, 2019c*) were obtained through the Human Protein Atlas (*Uhlén et al., 2015*).

## Statistics

Data are presented as mean ± standard error of the mean. Statistical significance was determined using an unpaired, two-tailed Student's T-test or one-way analysis of variance (ANOVA) followed by Tukey's multiple comparison test. Survival curves were generated with the Kaplan-Meier method and statistical comparisons were made using the log-rank method. p-values<0.05 were considered statistically significant. In vitro experiments were done in at least three biological replicates, which we define as parallel measurements of biologically distinct samples taken from independent experiments. Technical replicates we define as loading the same sample multiple times on the final assay. The in vivo experiments were done on at least five independent animals. Outliers were kept in the analyses, unless they were suspected to occur due to technical errors, in which case the experiment was repeated.

## Acknowledgements

This research was funded through grants obtained from the Swedish Cancer Foundation (Cancerfonden, CAN2017/518 and CAN2013/1273), The Swedish children's cancer foundation (Barncancerfonden), the Swedish society for medical research (SSMF, S17-0092), the OE och Edla Johanssons stiftelse and Olga Jönssons stiftelse. These funding sources were not involved in the study design; collection, analysis and interpretation of data; writing of the report; and in the decision to submit the article for publication. We would like to thank visiting students Kim Vanhollebeke and Justine Dobbelaere for their technical assistance; GradienTech for providing us with their CellDirector assays and Paul O´Callaghan for his valuable input on our project.

## Additional information

### Funding

| Funder | Grant reference number | Author |
| --- | --- | --- |
| Cancerfonden | CAN 2017/518 | Femke Heindryckx |
| Svenska Sällskapet för Medicinsk Forskning | S17-0092 | Femke Heindryckx |
| O. E. och Edla Johanssons Vetenskapliga Stiftelse | | Femke Heindryckx |
| Olga Jonssons stiftelse | | Femke Heindryckx |
| Cancerfonden | CAN2013/1273 | Femke Heindryckx |
| Barncancerfonden | | Pär Gerwins |

The funders had no role in study design, data collection and interpretation, or the decision to submit the work for publication.

### Author contributions

Nataša Pavlović, Formal analysis, Investigation, Methodology, Writing - original draft, Project administration, Writing - review and editing; Carlemi Calitz, Formal analysis, Investigation, Methodology, Writing - review and editing; Kess Thanapirom, Data curation, Investigation, Methodology; Guiseppe Mazza, Validation, Methodology; Krista Rombouts, Supervision, Investigation, Methodology, Project administration, Writing - review and editing; Pär Gerwins, Supervision, Funding acquisition, Project

administration, Writing - review and editing; Femke Heindryckx, Conceptualization, Resources, Data curation, Formal analysis, Supervision, Funding acquisition, Validation, Investigation, Visualization, Methodology, Writing - original draft, Project administration, Writing - review and editing

**Author ORCIDs**
Femke Heindryckx ⓘ https://orcid.org/0000-0002-1987-7676

**Ethics**
Animal experimentation: This study was performed in strict accordance with the recommendations by FELASA. All of the animals were handled according to approved institutional animal care and Uppsala University approved protocols were used. The protocol was approved by the Committee on the Ethics of Animal Experiments of Uppsala (C95/14). All effort was made to minimise suffering and to decrease animal usage.

**Decision letter and Author response**
Decision letter https://doi.org/10.7554/eLife.55865.sa1
Author response https://doi.org/10.7554/eLife.55865.sa2

## Additional files

### Supplementary files
• Supplementary file 1. Table with primer sequences.
• Supplementary file 2. Table with antibodies used for staining.
• Transparent reporting form

### Data availability
Proteomics data has been deposited in Dryad with the following DOI: https://doi.org/10.5061/dryad.6wwpzgmv2.

The following dataset was generated:

| Author(s) | Year | Dataset title | Dataset URL | Database and Identifier |
|---|---|---|---|---|
| Heindryckx F | 2020 | Protein expression of hepatocellular carcinoma in a fibrotic liver in mice | http://dx.doi.org/10.5061/dryad.6wwpzgmv2 | Dryad Digital Repository, 10.5061/dryad.6wwpzgmv2 |

The following previously published datasets were used:

| Author(s) | Year | Dataset title | Dataset URL | Database and Identifier |
|---|---|---|---|---|
| Seok JY, Na DC, Woo HG, Roncalli M, Kwon SM, Yoo JE, Ahn EY, Kim GI, Choi J, Kim YB, Park YN | 2020 | Fibrous stromal component in hepatocellular carcinoma reveals a cholangiocarcinoma-like gene expression trait and epithelial-mesenchymal transition | https://www.ncbi.nlm.nih.gov/geo/query/acc.cgi?acc=GSE31370 | NCBI Gene Expression Omnibus, GSE31370 |

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
