## [Decision Letter]

**Acceptance summary:**

The current manuscript reveals a previously unappreciated role for the unfolded protein response (UPR) in Hepatocellular carcinoma (HCC) progression. The authors demonstrate that factors, such as Transforming Growth Factor Β (TGFb), which are secreted by HCC cells cause the UPR concomitant with an activated phenotype in stellate cells. Activated stellate cells subsequently support HCC progression via an IRE1a-RNAse dependent mechanism. Accordingly, inhibiting IRE1a RNAse activity may be used to uncouple dynamic reciprocities between HCC and stromal stellate cells in order to mitigate neoplastic progression.

**Decision letter after peer review:**

Thank you for submitting your article "Inhibiting endoplasmic reticulum stress decreases tumor burden in a mouse model for hepatocellular carcinoma" for consideration by *eLife*. Your article has been reviewed by three peer reviewers, including Lynne-Marie Postovit as the Reviewing Editor and Reviewer #1, and the evaluation has been overseen by Päivi Ojala as the Senior Editor.

The reviewers have discussed the reviews with one another and the Reviewing Editor has drafted this decision to help you prepare a revised submission.

As the editors have judged that your manuscript is of interest, but as described below that additional experiments are required before it is published, we would like to draw your attention to changes in our revision policy that we have made in response to COVID-19 (https://elifesciences.org/articles/57162). First, because many researchers have temporarily lost access to their labs, we will give authors as much time as they need to submit revised manuscripts. We are also strongly recommending you to post the manuscript to bioRxiv (if it is not already there) along with this decision letter and a formal designation that the manuscript is 'in revision at *eLife*'. *eLife* is aiming to make posting to bioRxiv or medRxiv – either by the authors or the journal – the default for all *eLife* submissions. Authors will be able to opt out, but we will strongly encourage them not to. Please let us know if you would like to pursue this option. (If your work is more suitable for medRxiv, you will need to post the preprint yourself, as the mechanisms for us to do so are still in development.)

Summary:

The current manuscript by Pavlovic et al. focuses on the role of ER stress and UPR activation in communication between hepatocellular cancer cells (HCC) and hepatic stellate cells. This study builds upon a recent EMBO Molecular Medicine paper by the same group which reported a role for IRE1 in α-SMA activation in stellate cells treated with TGF-β. In the current study, the authors suggest that TGF-β from HCC cells causes the UPR in stellate cells and that these cells subsequently support pro-tumorigenic properties through an IRE1a-RNAse-dependent mechanism. While 4u8C (an IRE1alpha RNAse inhibitor) did seem to prevent pro-tumorigenic phenomena induced when HCC and LX2 stellate cells were co-cultured; the manuscript failed to convincingly demonstrate a role for HCC-derived TGF-β in the process. Moreover, mechanistic insight regarding how IRE1alpha RNAase activity in stellate cells may be promoting a pro-tumorigenic niche was lacking. Hence, the paper would require extensive revision before it would be acceptable for publications in *eLife*.

Essential revisions:

1) Many of the experiments related to the activation of the UPR were overly simplified with only certain aspects of each arm explored. For example, the data presented is limited to BiP, IRE1 and in some cases CHOP. To make such generalized statements referring to the UPR other arms PERK and ATF6 should also be included or the text amended to reflect that a full profile of UPR activation has not been completed.

2) The role of XBP-1 splicing / IRE1a-RNAse activity must be better substantiated, particularly since results often contradict each other, with variability in the effect of 4u8C. Validation or verification strategies must be included to verify the functionality of 4u8c inhibition, particularly in vivo. This could be done, for example, by assessing XBP1 splicing via QPCR or IHC in the DEN induced in vivo model. In addition, genetic approaches such as IRE1 knockdown or knockout are required to solidify those results obtained with 4u8c in the in vitro models, particularly given the potential for off-target effects (such as ROS generation) which may occur at the relatively high concentration of 4u8c (100uM) used in the current study. This was exasperated by the fact that 4u8c reduced phosphorylation of IRE1. In the original study Cross et al. (PNAS 2012) 32uM of 4u8c blocked XBP1 splicing in response to thapsigarin but did not alter IRE1 phosphorylation. Hence, the specificity of the P-IRE1 antibody used in this study should be validated.

3) The results in Figure 4—figure supplement 1 actually refute the hypothesis that HCC-derived TGFβ may be promoting the UPR in stellate cells. In fact, the results rather suggest that an autocrine signaling mechanism may be playing a role in the LX2 cells and that the HepG2 cells may even prevent this somehow. Accordingly, another mechanism of action should be considered using genetic approaches in the stellate versus HCC cells and/or the conclusions drawn from this work should be altered.

4) The quality of the microscopy images presented could in many instances be improved. The co-localization referred to by the authors in many figures e.g. Supplemental Figure 1D and E is not convincing. In Supplemental Figure 1 E, the staining looks to be in the pericytes. Please include more pictures and/or ensure cells are stellate cells.

[Editors' note: further revisions were suggested prior to acceptance, as described below.]

Thank you for submitting your article "Inhibiting IRE1α-endonuclease activity decreases tumor burden in a mouse model for hepatocellular carcinoma." for consideration by *eLife*. Your article has been reviewed by two peer reviewers, and the evaluation has been overseen by a Reviewing Editor and Päivi Ojala as the Senior Editor. The reviewers have opted to remain anonymous.

The reviewers have discussed the reviews with one another and the Reviewing Editor has drafted this decision to help you prepare a revised submission.

Summary:

The revised manuscript has been looked at again by the reviewers. While the paper has been improved, a number of concerns remain, which I believe are significant enough to warrant further revisions.

Significant concerns remain regarding the of target effect of 4u8c. While we agree that IRE1 signalling itself could be triggering ROS, it would be prudent to ensure that the observations attributed to 4u8c in this study are due to its on-target effect on IRE1 and not its off-target antioxidant effect as has previously been reported. The authors should add 4u8c to IRE1 knockout cells, at least in the HepG2 cell line wherein it seems that ROS levels are deferentially affected by 4u8c and IRE1 knockdown.

This revised panels in Figure 2 and Figure 2—figure supplement 1 do not convincingly demonstrate co-localisation with XBP1s. Specifically, the level of signal for IRE1, p-IRE1 and BiP is very low, making it difficult to formulate a conclusion. Please improve these panels.

Please correct labelling and include molecular weight markers in Figure 2E. As it stands, the molecular weights of the bands indicated on the Western blot do not correspond to XBP1 unspliced (which should be 33kDa) and spliced (which should be 55 kDa).

In Figure 2H, XBP1s localisation seems to be entirely cytoplasmic. This is rather unexpected as at least some XBP1s would be expected to localise to the nucleus. Please comment on this result in relation to the extant literature.

In Figure 4, lanes 2 and 4 have no product, despite the fact that Pst1 should digest XBP1 unspliced but not XBP1 spliced. The authors should carefully check and/or comment upon this result.

Please confirm IRE1 knockdown at the protein level.

---

## [Author Response]

Essential revisions:1) Many of the experiments related to the activation of the UPR were overly simplified with only certain aspects of each arm explored. For example, the data presented is limited to BiP, IRE1 and in some cases CHOP. To make such generalized statements referring to the UPR other arms PERK and ATF6 should also be included or the text amended to reflect that a full profile of UPR activation has not been completed.

We completely agree that this manuscript mainly focusses on the IRE1a branch of the UPR. We have therefore updated the text, so that we do not make generalized statements regarding ER-stress, but specify that the effects we see is through IRE1a. In addition, we have also presented more data regarding the involvement of PERK and ATF6 in the animal model (Figure 2A) and the co-cultures (Figure 4A).

2) The role of XBP-1 splicing / IRE1a-RNAse activity must be better substantiated, particularly since results often contradict each other, with variability in the effect of 4u8C. Validation or verification strategies must be included to verify the functionality of 4u8c inhibition, particularly in vivo. This could be done, for example, by assessing XBP1 splicing via QPCR or IHC in the DEN induced in vivo model.

Thank you for this suggestion. It is important to note that we have used a concentration of 4u8C that has been shown to be effective in several other in vivo studies [1-3]. We have now looked at the effect of 4u8C via PCR + enzymatic digestion with Pst-I, Western blot and immunohistochemical staining (Figure 2D-I). This shows that 4u8C-treatment decreases splicing of XBP1 in our mouse model.

In addition, genetic approaches such as IRE1 knockdown or knockout are required to solidify those results obtained with 4u8c in the in vitro models, particularly given the potential for off-target effects (such as ROS generation) which may occur at the relatively high concentration of 4u8c (100uM) used in the current study. This was exasperated by the fact that 4u8c reduced phosphorylation of IRE1. In the original study Cross et al. (PNAS 2012) 32uM of 4u8c blocked XBP1 splicing in response to thapsigarin but did not alter IRE1 phosphorylation. Hence, the specificity of the P-IRE1 antibody used in this study should be validated.

The concentration of 4u8C that we used are within the range of what is described in literature [1-4]. In the original study from Cross et al., no reduction in p-IRE1a was seen after treatment with 4u8C [5]. However, subsequent studies have also shown that 4u8C treatment can reduce IRE1a-phosphorylation. The Nelson et al. (J. Immunol, 2018) study described 4u8C as an inhibitor of IRE1a-phosphorylation in B cells [6, 7]. Similarly, treatment with 4u8C reduced the protein levels of p-IRE1a in Hela cells [8] and in tunicamycin-induced cardiomyocytes [9].

To assess if the effect is through generation of ROS, we have quantified the intracellular ROS levels after treatment with 50uM and 100 μm of 4u8C (Figure 10). After 6 hours, 4u8C decreased the generation of ROS in LX2, HepG2 and Huh7 cells in both concentrations. This could mean that the results could be partially caused by decreasing ROS-generation. This is in line with the study from Chan et al. (2018), who describes that 4u8C is a potent ROS-scavenger [5]. However, studies have shown that IRE1a itself plays an important role in the generation of ROS [10]. IRE1α generates ROS through Ca^2+^- mediated signaling between the IRE1α-InsP3R pathway in the ER and the redox-dependent apoptotic pathway in the mitochondrion, as well as via induction of activation of CHOP, BiP and through spliced XBP1 [10-12]. Therefore, the relation between IRE1a reduction with 4u8C and the effect on ROS-generation might be more complicated. To support this, we transfected LX2-cells, HepG2-cells and Huh7-cells with si-IRE1a. Interestingly, a similar reduction in ROS generation was seen after transfection of LX2-cells as in the 4u8C-treated LX2-cells, while an increase of ROS generation was noted in the transfected HepG2-cells. These results indicate that the reduction in ROS could partially be explained through the decreased activation of the IRE1a-pathway in the LX2-cells. However, how the IRE1a-pathway affects the generation of ROS, seems to be cell-type dependent, as we see different results in the different cell lines we tested. This is in line with previous studies, which also observed this cell-line dependent effect on IRE1a-dependent ROS-generation.

To further solidify our results obtained with 4u8C, we have done transfections in the in vitro models (Figure 9 and Figure 9—figure supplement 1) using an siRNA that silences IRE1a in the hepatic stellate cells and tumor cells. Silencing IRE1a in the LX2 cells significantly decreases mRNA expression of proliferation marker PCNA, as well as the relative cell number measured by resazurin reduction assay in co-cultures. Similarly, silencing IRE1a in the HepG2 cells decreased proliferation by 45% in the monocultures and 30% in the co-cultures. In contrast to this, we saw an increase of 47% and 27% after silencing IRE1a in the Huh7 monocultures and co-cultures. We have now changed the conclusions of the manuscript so that we not only focus on the effect on stellate cells, but also explain that part of the results can be explained through a direct effect on tumor cells, which seems to be cell-line specific.

To test whether these differences between the cell lines could be attributed to differences in ROS-generation, we measured ROS using DCFDA (Figure 10). We observed that transfecting the different cell lines with si-IRE1a affected the generation of ROS, but the extend and direction of this effect seemed to be cell line dependent. Oxidative stress can affect cell proliferation, specifically in cancer cells and stellate cells. In addition, ROS is one of the critical mediators of stellate cell activation and ECM-production. Therefore, our results can also be explained through IRE1a-dependent mechanisms of ROS-generation. Further studies are needed to verify this interesting link between IRE1a and ROS-generation, however, we feel like this is beyond the scope of our current study. This has now been added to the Discussion.

3) The results in Figure 4—figure supplement 1 actually refute the hypothesis that HCC-derived TGFβ may be promoting the UPR in stellate cells. In fact, the results rather suggest that an autocrine signaling mechanism may be playing a role in the LX2 cells and that the HepG2 cells may even prevent this somehow. Accordingly, another mechanism of action should be considered using genetic approaches in the stellate versus HCC cells and/or the conclusions drawn from this work should be altered.

We have now added additional data showing the effect of IRE1a-silencing in the HCC-cells compared to the silencing in the LX2-cells, provide an alternative mechanism through the effect of IRE1a-inhibition on ROS-production and also mention other alternative mechanisms in the Discussion.

4) The quality of the microscopy images presented could in many instances be improved. The co-localization referred to by the authors in many figures e.g. Supplemental Figure 1D and E is not convincing. In Supplemental Figure 1 E, the staining looks to be in the pericytes. Please include more pictures and/or ensure cells are stellate cells.

We have included more images to the manuscript (Figure 2—figure supplement 1) and also clarify that there is some co-localization, but that the expression is not exclusively located in the hepatic stellate cells. It is also important to note that hepatic stellate cells are liver-specific pericytes, that reside in the perisinusoidal space of Disse, between endothelial cells and hepatocytes.

[Editors' note: further revisions were suggested prior to acceptance, as described below.]

Revisions for this paper:Significant concerns remain regarding the of target effect of 4u8c. While we agree that IRE1 signalling itself could be triggering ROS, it would be prudent to ensure that the observations attributed to 4u8c in this study are due to its on-target effect on IRE1 and not its off-target antioxidant effect as has previously been reported. The authors should add 4u8c to IRE1 knockout cells, at least in the HepG2 cell line wherein it seems that ROS levels are deferentially affected by 4u8c and IRE1 knockdown.

For the revised version of this manuscript, we have added the requested experiment and we indeed see that 4u8C continues to decrease ROS-production in the transfected cells. We agree that part of the results could be explained through an off-target effect and more experiments are needed to assess this possibility. Therefore, we have altered our claims in the Discussion and specify that an off-target effect is also a possible explanation for some of the conclusions drawn in this paper. We are continuing our studies on the effect of 4u8C as a ROS-scavenger, as well as the potential interaction between the IRE1a-pathway and ROS-generation. In a preliminary experiment, we found that treatment with the ER-stress inducer tunicamycin, increases the levels of ROS in HepG2 and Huh7 cell lines, thus possibly supporting our hypothesis that there is an interaction between ER-stress pathways and ROS-generation (Author response image 1). In this experiment we did not observe the same markable decrease in ROS-production after 4u8C-treatment, however, it is important to note that this was measured after a 2 hour incubation time, while in the manuscript we used 6 hours. Treatment with 4u8C seems to affect ROS-generation in a time-dependent matter, so this would probably explain the discrepancy (Author response image 1).

**Author response image 1. sa2fig1:** Effect of ER-stress on ROS production. (A) the ER-stress inducer increases ROS levels in HepG2 and Huh7 cells. (B) The ROS-scavenging effect of 4u8C increases over time.

In our ongoing studies, we also found co-culturing cells in different combinations with an IRE1a-silenced cell line affects ROS-levels and that this effect is different compared to mono-cultured cell lines (Author response image 2). The most remarkable finding from this experiment is that transfecting LX2-cells with si-IRE1a increases ROS-generation in the LX2 + HepG2 co-culture, while a decrease is seen in the LX2-monoculture (Author response image 2). More studies are necessary to verify what this means in terms of explaining the interaction between stellate cells and tumor cells. The generation of ROS is a very well-known contributor to stellate cell activation in chronic liver disease, as infiltrated inflammatory cells and Kupffer cells produce ROS to induce activation and proliferation of stellate cells. However, ROS-generation can also have the opposite effect, as exogenous and intracellular oxidative stress can induce cell death of activated stellate cells, thereby contributing to the resolution of fibrosis. We hope to study the role of IRE1a-mediated ROS-generation in the interaction between stellate cells and tumor cells in the future, either by adding additional information to this existing paper (through BioRxiv or *eLife*´s Research Advance) or publish this as a follow-up paper.

**Author response image 2. sa2fig2:** Effect of silencing IRE1a in different co-culture conditions. (A) LX2-cells transfected with si-RNA targeting IRE1a or mock-transfected (Scr) in mono-culture or co-culture with tumor cells (HepG2 and Huh7). (B) HepG2 or Huh7-cells transfected with si-RNA targeting IRE1a or mock-transfected (Scr) in mono-culture or co-culture with LX2-stellate cells.

This revised panels in Figure 2 and Figure 2—figure supplement 1 do not convincingly demonstrate co-localisation with XBP1s. Specifically, the level of signal for IRE1, p-IRE1 and BiP is very low, making it difficult to formulate a conclusion. Please improve these panels.

We have now added another supplementary figure, where we provide images at a higher magnification. We would like to point out that we do not use the term “co-localization” as it is unlikely that there would be expression of XBP1s and aSMA within the same pixel. What we state in the text is that expression of aSMA and several ER-stress markers is localized within close proximity from each other and that it is probably localized within the same cell, especially when considering the location of the nearby nuclei on Hoechst staining. We hope that the higher magnification clarifies this issue.

Please correct labelling and include molecular weight markers in Figure 2E. As it stands, the molecular weights of the bands indicated on the Western blot do not correspond to XBP1 unspliced (which should be 33kDa) and spliced (which should be 55 kDa).

Thank you for pointing this out, we have now corrected this in the figure.

In Figure 2H, XBP1s localisation seems to be entirely cytoplasmic. This is rather unexpected as at least some XBP1s would be expected to localise to the nucleus. Please comment on this result in relation to the extant literature.

Indeed, we do not observe a clear nuclear expression of spliced XBP1, which is in contrast to the study of Yoshida et al., which shows that spliced XBP1 predominantly localizes in the nucleus of HeLa-cells exposed to acute ER-stress [13]. This study also describes a mechanism whereby unspliced-XBP1 forms a complex with the spliced isoform, thereby exporting it from the nucleus to the cytoplasm, resulting in subsequent degradation by the proteasome. However, this event has been described during the recovery phase of an acute ER-stress event. In our mouse model, we have treated the mice with a carcinogenic compound for 25 weeks, which results in a chronic inflammation and a subsequent activation of the ER-stress pathways. It is therefore not unlikely that different cells in this model are experiencing different phases of acute ER-stress and recovery. At a higher magnification, it becomes clear that the expression of spliced XBP1 is not only cytoplasmic but some staining appears peri-nuclear and nuclear. This could represent different stages of ER-stress activation and recovery in different cell populations; however, more experiments would be needed to verify this hypothesis. A summarized version of this explanation has been added to the Discussion.

In Figure 4, lanes 2 and 4 have no product, despite the fact that Pst1 should digest XBP1 unspliced but not XBP1 spliced. The authors should carefully check and/or comment upon this result.

We are aware of that there seems to be no product in lanes 2 and 4 in the image provided in the manuscript. We have first checked several issues to verify that this is not due to a technical artifact. Firstly, all conditions contain the same amount of cDNA. This cDNA has also been used for qPCR, where all samples have a stable expression the reference genes. Secondly, in other biological replicates of this experiment we did observe a faint band in these lanes (Author response image 3). Thirdly, when looking at the raw qPCR-data from spliced and unspliced XBP1, we also note that in samples from these conditions have a very low expression of unspliced XBP1, with CT values between 30 and 34, while GAPDH comes up around cycle 20. Therefore, we suspect that this is due to a low expression of total XBP1 (spliced and unspliced). Discrepancies between the baseline expressions of XBP1 can be observed between different studies, depending on the cell type or tissue used. For instance, the study of Cassimeris et al., describes barely detectable levels of both spliced and unspliced XBP1 in epidermal keratinocytes of the healthy hind limbs of animals with endocrinopathic equine laminitis, while a high level of total XBP1 was observed in these cells derived from the diseased front limbs of the same animal [14]. These levels correlated with the overall levels of ER-stress, which differed between diseased and healthy tissue and thereby suggest that both unspliced and spliced XBP1 can be affected by levels of ER-stress. In line with our findings, the study of Kishino et al. (2017) reports constitutively low levels of unspliced XBP1 in untreated auditory cells, which increases during tunicamycin-induced ER-stress. In this study, the expression level of unspliced XBP1 peaked at 24 h and decreased at 48 h after the treatment with tunicamycin, thus suggesting that levels of unspliced XBP1 can also be affected by the levels of ER-stress in a time-dependent matter [15]. The conditions where we observe lower levels of both unspliced and spliced XBP1 correspond to those where we expect to see generally low levels of IRE1a activation, namely LX2-monocultures and LX2+Hep2 co-cultures treated with 4u8C. A summarized version of this explanation has been added to the Discussion.

**Author response image 3. sa2fig3:** Visible bands in lane 2 and 4, which corresponds to LX2-monocultures and LX2+Hep2 co-cultures treated with 4u8C.

Please confirm IRE1 knockdown at the protein level.

We did not have an optimized protocol for western blot using the IRE1a antibody that we use in our lab. During one of our first tests, we managed to see a protein band for IRE1a in the Huh7-cells, clearly showing that the si-RNA decreased protein levels of IRE1a (Author response image 4). Unfortunately, we have used up all the si-RNA reagents at the moment and cannot repeat this experiment on the LX2 and HepG2 cells. A new order has been placed, but due to the ongoing COVID19 pandemic both the si-RNA and the Ingenio electroporation solution are currently back-ordered. We understand that this experiment is an important part of *eLife*´s quality control, however, we hope that the editor and reviewers are understanding about the current unusual conditions and that we can submit this information at a later time-point through biorxiv.

**Author response image 4. sa2fig4:** Protein levels of IRE1a in mock transfected and si-IRE1a transfected Huh7 cells.

**References:**

1. Heindryckx, F., et al., Endoplasmic reticulum stress enhances fibrosis through IRE1alpha-mediated degradation of miR-150 and XBP-1 splicing. EMBO Mol Med, 2016. 8(7): p. 729-44.2. Qiu, Q., et al., Toll-like receptor-mediated IRE1alpha activation as a therapeutic target for inflammatory arthritis. EMBO J, 2013. 32(18): p. 2477-90.3. Lebeaupin, C., et al., Bax inhibitor-1 protects from nonalcoholic steatohepatitis by limiting inositol-requiring enzyme 1 α signaling in mice. Hepatology, 2018. 68(2): p. 515-532.4. Stewart, C., et al., Regulation of IRE1alpha by the small molecule inhibitor 4mu8c in hepatoma cells. Endoplasmic Reticulum Stress Dis, 2017. 4(1): p. 1-10.5. Chan, S.M.H., et al., The inositol-requiring enzyme 1 (IRE1alpha) RNAse inhibitor, 4micro8C, is also a potent cellular antioxidant. Biochem J, 2018. 475(5): p. 923-929.6. Nelson, A.M., et al., RNA Splicing in the Transition from B Cells to Antibody-Secreting Cells: The Influences of ELL2, Small Nuclear RNA, and Endoplasmic Reticulum Stress. J Immunol, 2018. 201(10): p. 3073-3083.7. Carew, N.T., et al., Linking Endoplasmic Reticular Stress and Alternative Splicing. Int J Mol Sci, 2018. 19(12).8. Li, Y., et al., eIF2alpha-CHOP-BCl^-^2/JNK and IRE1alpha-XBP1/JNK signaling promote apoptosis and inflammation and support the proliferation of Newcastle disease virus. Cell Death Dis, 2019. 10(12): p. 891.9. Liu, M., et al., Activation of the unfolded protein response downregulates cardiac ion channels in human induced pluripotent stem cell-derived cardiomyocytes. J Mol Cell Cardiol, 2018. 117: p. 62-71.10. Zeeshan, H.M., et al., Endoplasmic Reticulum Stress and Associated ROS. Int J Mol Sci, 2016. 17(3): p. 327.11. Abuaita, B.H., et al., The Endoplasmic Reticulum Stress Sensor Inositol-Requiring Enzyme 1alpha Augments Bacterial Killing through Sustained Oxidant Production. mBio, 2015. 6(4): p. e00705.12. Riaz, T.A., et al., Role of Endoplasmic Reticulum Stress Sensor IRE1alpha in Cellular Physiology, Calcium, ROS Signaling, and Metaflammation. Cells, 2020. 9(5).13. Yoshida, H.; Oku, M.; Suzuki, M.; Mori, K., pXBP1(U) encoded in XBP1 pre-mRNA negatively regulates unfolded protein response activator pXBP1(S) in mammalian ER stress response. J Cell Biol 2006, 172, (4), 565-75.14. Cassimeris, L.; Engiles, J. B.; Galantino-Homer, H., Detection of endoplasmic reticulum stress and the unfolded protein response in naturally-occurring endocrinopathic equine laminitis. BMC Vet Res 2019, 15, (1), 24.15. Kishino, A.; Hayashi, K.; Hidai, C.; Masuda, T.; Nomura, Y.; Oshima, T., XBP1-FoxO1 interaction regulates ER stress-induced autophagy in auditory cells. Sci Rep 2017, 7, (1), 4442.